# A tuneable minimal cell membrane reveals that two lipid species suffice for life

Isaac Justice [1], Petra Kiesel[2], Nataliya Safronova [1], Alexander von Appen [2] & James P. Saenz [1,3] ✉

All cells are encapsulated by a lipid membrane that facilitates their interactions with the environment. How cells manage diverse mixtures of lipids, which dictate membrane property and function, is experimentally challenging to address. Here, we present an approach to tune and minimize membrane lipid composition in the bacterium *Mycoplasma mycoides* and its derived 'minimal cell' (JCVI-Syn3A), revealing that a two-component lipidome can support life. Systematic reintroduction of phospholipids with different features demonstrates that acyl chain diversity is more important for growth than head group diversity. By tuning lipid chirality, we explore the lipid divide between Archaea and the rest of life, showing that ancestral lipidomes could have been heterochiral. However, in these simple organisms, heterochirality leads to impaired cellular fitness. Thus, our approach offers a tunable minimal membrane system to explore the fundamental lipidomic requirements for life, thereby extending the concept of minimal life from the genome to the lipidome.

Cell membranes are complex and responsive systems that serve to protect and mediate interactions of life with its environment. A large part of this molecular complexity is due to the diverse panel of lipids that make up the lipidome and which ultimately determine the form and function of the membrane. The complexity of cellular lipidomes can be staggering, from tens of unique structures in bacteria[1], to hundreds in eukaryotic organisms[2]. How life has evolved to utilize such complex mixtures of lipids to build cellular membranes remains an active area of exploration[1,3–6]. While synthetic membranes can be constructed using a single lipid species, the minimal number of lipid species required for a functional cell membrane remains undetermined. Identifying a minimal viable lipidome would provide a critical starting point for elucidating the combinations of lipid structures that are essential for membrane integrity and function, offering insights into the fundamental chemical and physical requirements of cellular life.

One approach to studying lipidome complexity is to experimentally manipulate lipidome composition and observe effects on cell fitness (e.g. growth). Bacterial model organisms have proven to be excellent systems for tuning lipidome composition by genetically disrupting lipid biosynthesis[7]. For example, early work with *Escherichia coli* mutants deficient in fatty acid synthesis explored the role of acyl chain unsaturation for cell growth[8], and a similar approach with *Bacillus subtilis* mutants explored the role of branched acyl chains[9]. More recent work, employing approaches to tune lipid unsaturation, revealed the importance of homeoviscous adaptation for electron transport[10] and demonstrated how low membrane fluidity can induce phase separation, impaired membrane potential, cell growth and division[11]. However, for interpreting how lipidome composition and flexibility are affected by perturbed lipid synthesis, *E. coli*, like many bacteria, is complicated by the fact that it has multiple membranes (e.g. inner and outer). Changes in a single membrane are obfuscated by whole cell lipid extracts, and purifying specific membrane types is laborious, and can lead to substantial experimental error due to varying purity. Gram-positive organisms such as *B. subtilis*, that have only a single membrane are better models in this regard[12,13]. However,

[1]Technische Universität Dresden, B CUBE Center for Molecular Bioengineering, Dresden, Germany. [2]Max Planck Institute of Molecular Cell Biology and Genetics, Pfotenhauerstrasse 107, Dresden, Germany. [3]Technische Universität Dresden, Faculty of Medicine, Dresden, Germany. ✉e-mail: james.saenz@tu-dresden.de

traditional genetic approaches to tuning lipidome composition have so far not afforded complete control over both lipid class composition, and phospholipid acyl chain composition. Thus, a cellular model system in which lipidome complexity can be reduced in a systematic fashion has not yet been established.

*Mycoplasmas* are a genus of genomically simple bacterial pathogens possessing several features that are promising as tunable living model membrane systems. *Mycoplasmas* have a single plasma membrane, and lack a cell wall[14]. Therefore, their membrane composition and biophysical properties can be examined in situ without the need for laborious membrane isolation. Having evolved to a parasitic lifestyle, *Mycoplasmas* have lost many pathways for biomolecular synthesis, including most of their lipid synthesis pathways, relying instead on lipids acquired from their hosts or from the growth media[14–18]. This affords the possibility to control their lipidome composition by controlling what lipids are provided in the media. Further, because of their relatively small genomes, the number of components involved in managing lipidome composition and membrane adaptation is within reach of being completely characterized and modeled[19]. Recently, a minimal cell, JCVI-Syn3.0, was engineered from *Mycoplasma mycoides subsp. capri* strain GM12 by systematically removing genes from *M. mycoides* to achieve an organism where every remaining gene is essential or quasi-essential[20]. However, this organism exhibited pleomorphism and irregular cell division[21]. To restore normal cell division, a new strain, JCVI-Syn3A was created with 19 additional genes not present in JCVI-Syn3.0. The addition of these genes resulted in a regularly dividing quasi-minimal cell which offers a platform to study the role of lipids for the most fundamental requirements of life in a genomically minimized system.

In this study, we establish an approach to modulate lipidome composition and reduce its complexity in *M. mycoides* and JCVI-Syn3A. By tuning the lipid composition of the growth medium and introducing diether phospholipids we introduce an approach to bypass cellular lipid remodeling, achieving a lipidome with only two lipids. Using these lipidomically minimal living membranes, we compare the relative importance of phospholipid head group vs. acyl chain complexity. Additionally, we observed profound effects resulting from subtle changes in the lipid chirality that distinguishes Archaea from the rest of life. By developing approaches to tune and minimize mycoplasma lipidome composition, we hope to introduce a new tool for deciphering the principles of living membranes and to introduce a new paradigm for understanding why life has evolved to utilize so many lipids.

## Results

### Mycoplasmas as minimal model membrane systems

*Mycoplasma mycoides* is a pathogen of mammals that has historically been used as a simple model membrane system. There are a number of reasons why *M. mycoides* makes a good model membrane system. First, its small genome (~1,100,000 bp) limits the complexity of its genetic regulation, and has allowed researchers to fully sequence and annotate the genome (although a number of genes still have unknown or only putative functional assignments)[22]. Second, as a pathogen, the primary source of lipids for *M. mycoides* is through environmental uptake from the host rather than synthesis[14–16,23]. In a laboratory setting using growth medium as the lipid source, this feature provides the experimenter with direct control over the lipid components available for synthesizing and maintaining the membrane; by adding or removing specific lipids from the growth medium of *M. mycoides*, the composition of the membrane can be altered. Third, *M. mycoides* has a single plasma membrane and no cell wall or organelles, making it easy to interrogate the membrane and ensuring that membrane targeting probes in *M. mycoides* are acting on the membrane and not another structure[14]. Fourth, and perhaps most importantly for this research, *M. mycoides* is unable to synthesize or alter fatty acid composition[14–17].

This means that, while it can alter the acyl chain composition of its membrane lipids, it is limited in its ability to do so by the pool of fatty acids it has access to. These factors combine to make *M. mycoides* one of the model systems in which membrane remodeling is both the simplest and most controllable, and as such it is an excellent model system to study membrane remodeling in living organisms[24].

In *M. mycoides* membranes phospholipids, sterols, and free fatty acids are taken up from the environment and either incorporated into the membrane or taken up into the cytoplasm (Fig. 1(1))[14,25–27]. *M. mycoides* can cleave acyl chains from exogenous phospholipids (Figs. 1(2))[28–30]. When a pool of free fatty acids is present, *M. mycoides* can use those fatty acids to modify the acyl chain composition of phospholipids taken up from the media, or can synthesize the phosphatidylglycerol (PG) class of lipids which can in turn be modified and used to synthesize cardiolipin (CL) (Figs. 1(3–5))[14,27,28,31–34]. These newly synthesized or modified lipids can be broken down to replenish the fatty acid pool, or reinserted into the membrane (Fig. 1(6, 7))[32]. With the exception of the modification of PG headgroups to make cardiolipin, all lipid remodeling in *M. mycoides* is acyl chain remodeling–that is, it relies on having access to a pool of free fatty acids to modify the existing acyl chain composition of phospholipids or synthesize de novo PGs[14,15]. Cholesterol is essential for growth of *M. mycoides* (Fig. 1(8))[26,35,36]. For the first time, we report the feeding of *M. mycoides* on a defined lipid diet which contains no exogenous source of free fatty acids. When fed such a diet, *M. mycoides* is forced to rely completely on acyl chain scavenging from exogenous phospholipids as its only source of acyl chains for which to remodel its membrane lipidome.

The diverse lipid structures that *M. mycoides* can take up or synthesize ultimately determine the physical properties of their cell membranes[37,38]. Phospholipid acyl chains can vary in terms of their length and degree of unsaturation, both influencing physical parameters such as membrane fluidity, thickness, and permeability. Phospholipid headgroups such as PG and cardiolipin introduce a negative surface charge to the membrane, which can influence their interaction with peripheral membrane proteins. Conversely, PC is zwitterionic and introduces a neutral charge to the membrane surface. The geometric shape of phospholipids is also important, and determines whether a lipid spontaneously aggregates to form a bilayer or non-bilayer structure. For example, cardiolipin has four acyl chains and a relatively small headgroup giving it a conical profile. Cells have been shown to tune the abundance of such conical lipids to modulate the curvature and bending rigidity of their membranes[39]. Sterols, which also do not form bilayers by themselves, play an important biophysical role in the membrane, including in modulating membrane fluidity, stability, facilitating liquid-liquid phase separation, and membrane asymmetry[40,41]. By limiting the diversity of lipids that can be taken up or synthesized, we aimed to identify a minimal viable lipidome that can be used as an experimental platform in which lipid diversity can be systematically tuned in a living membrane.

### Minimizing the lipidome

Our first goal was to determine the minimal lipidome that can support growth of *M. mycoides*. Sterols, preferably cholesterol, are required for growth, and must be included in any lipid diet. *M. mycoides* can synthesize several phospholipids (e.g. PG and cardiolipin; full lipid names and abbreviations can be found in Table 1) when provided free fatty acids[14,27] (Fig. 2a,b). For example, when grown on a lipid diet consisting of cholesterol, and two fatty acids (palmitate - C16:0 and oleate - C18:1), the lipidome of *M. mycoides* contains primarily cholesterol, PG, cardiolipin and small amounts of diacylglycerol (DAG) and phosphatidic acid (PA), both precursors of PG synthesis (Fig. 2b, Supplementary Data S3). Therefore, to minimize the phospholipid diversity, we removed free fatty acids from the lipid diet and instead provided a single phospholipid, 16:0/18:1 phosphatidylcholine (POPC), along with

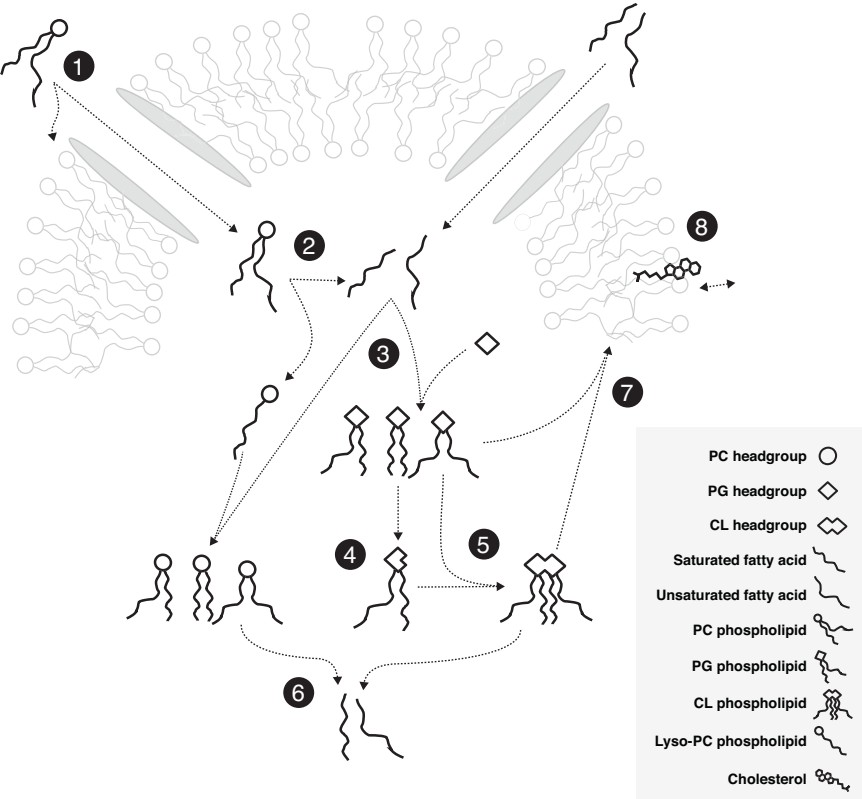

**Fig. 1 | Membrane remodeling in *M. mycoides* is dependent on acyl chain scavenging. 1** *Mycoplasma mycoides* can take up phospholipids and free fatty acids from its environment. **2** Cleaving ester bonds from glycerophospholipids results in a pool of free fatty acids. **3** With fatty acids, *M. mycoides* is able to synthesize phosphatidylglycerol (PG) with various fatty acid compositions, as well as remodel

other phospholipid classes. **4** Cardiolipin (CL) can be synthesized from two PGs, starting with cleavage of PG headgroups. **5** *M. mycoides* can synthesize a variety of cardiolipins from PG. **6** Cleaving acyl chains from remodeled and synthesized phospholipids replenishes the pool of free fatty acids. **7** Glycerophospholipids are inserted into the membrane. **8** Cholesterol is essential for *M. mycoides* growth.

cholesterol, yielding a minimal 2-component lipid diet. When transferred from growth medium containing fetal bovine serum (FBS, a complex undefined lipid source), to the two-component lipid diet, cells initially grew poorly and tended to aggregate in clumps (not shown). Thin layer chromatography (TLC) of cell lipid extracts showed that, initially, PG and cardiolipin (both internally synthesized lipids) disappeared, and only traces of sphingomyelin (SM) remained, presumably carryover from growth on FBS (Fig. 2a; full TLCs are shown in Supplementary Fig. S2). After adaptation (> 3 passages in batch culture), SM was no longer visible. However, phosphatidylglycerol (PG) and cardiolipin reappeared along with lyso-PC (a PC lipid with one of its acyl chains removed, which appears below the SM band). Since *M. mycoides* cannot synthesize phospholipids in the absence of fatty acids, these observations indicate that acyl chains were being scavenged from POPC for the synthesis of PG and cardiolipin, presumably as a result of lipase activity (Fig. 2a). A systematic analysis of PG and cardiolipin production on minimal lipid diets composed of varying phospholipid species demonstrated that *M. mycoides* is capable of scavenging acyl chains from a broad range of phospholipid headgroups (Fig. S1, Supplementary Data S2), demonstrating a robust capacity for *M. mycoides* to procure acyl chains for internal PG and cardiolipin synthesis from nearly any phospholipid source.

The capacity for *M. mycoides* to scavenge acyl chains from exogenous phospholipids provided a hurdle to our goal of minimizing the lipidome. Shotgun mass spectrometry of cells grown on the minimal 2-component lipid diet revealed 31 lipid species generated from the remodeling of POPC (with 18 lipid species comprising 99% of the lipidome), and presence of phospholipids including PG, cardiolipin, with small amounts of the PG precursors PA and DAG (Fig. 2c,

Supplementary Data S3). Since the source of acyl chains for internal phospholipid synthesis presumably came from exogenous POPC, we reasoned that internal lipid synthesis could be eliminated by blocking acyl chain scavenging. To do this, we replaced POPC with an analogous lipid containing ether-linked 16:0 and 18:1 hydrocarbon chains, which are inert to lipase activity[42,43]. Following the transfer of cells from FBS or minimal POPC +cholesterol diet to a minimal diether PC (D.PC) +cholesterol diet, TLC analysis of lipid extracts over several passages showed the disappearance of cardiolipin and PG, and presence of only two bands corresponding to cholesterol and D.PC (Fig. 2a). Shotgun lipidomic analysis demonstrated that cholesterol and D.PC accounted for 99.9 mol% of the detected lipids (Fig. 2d) (the remaining very low abundance lipids are derived from impurities in the media, in particular from yeast extract, Supplementary Data S3). Thus, by introducing an enzymatically inert phospholipid, D.PC, to the minimal lipid diet, we could achieve a minimal lipidome composed predominantly of only two lipid species, cholesterol and D.PC.

## Tuning lipidome composition

Minimizing the lipidome of *M. mycoides* to two lipids resulted in a two-fold decrease in growth rates (Fig. 3a, D.PC condition compared to POPC condition). We next asked which components of the lipidome most affect growth. By feeding cells with D.PC, we had reduced lipidome composition in several ways. First, we eliminated the presence of internally synthesized phospholipids such as PG and cardiolipin. Therefore, minimizing the diversity of phospholipid headgroups could have contributed to impaired growth. Second, we eliminated phospholipid hydrocarbon chain diversity by limiting the cell to one configuration (16:0/18:1). Thus, it is also possible that restricting the

**Table 1 | Lipid names and abbreviations**

| Lipid class name | Lipid class abbreviation | Lipid species name | Lipid Species Abbreviation |
|---|---|---|---|
| Sterol | N/A | Cholesterol | N/A |
| Diether Phosphatidylcholine | diether PC | 1-O-hexadecanyl-2-O-(9Z-octadecenyl)-sn-glycero-3-phosphocholine | diether PC |
| Diether Phosphatidylglycerol | diether PG | 1-O-hexadecanyl-2-O-(9Z-octadecenyl)-sn-glycero-3-phospho-(1'-rac-glycerol) | diether PG |
| Diether Phosphatidylethanolamine | diether PE | 1-O-hexadecanyl-2-O-(9Z-octadecenyl)-sn-glycero-3-phosphoethanolamine | diether PE |
| Phosphatidylcholine | PC | 1-palmitoyl-2-oleoyl-glycero-3-phosphocholine | POPC |
| | | 1,2-dipalmitoyl-sn-glycero-3-phosphocholine | DPPC |
| | | 1,2-dioleoyl-sn-glycero-3-phosphocholine | DOPC |
| Phosphatidylglycerol | PG | 1-palmitoyl-2-oleoyl-sn-glycero-3-phospho-(1'-rac-glycerol) | POPG |
| | | 1,2-dipalmitoyl-sn-glycero-3-phospho-(1'-rac-glycerol) | DPPG |
| | | 1,2-dioleoyl-sn-glycero-3-phospho-(1'-rac-glycerol) | DOPG |
| Phosphatidylethanolamine | PE | 1-palmitoyl-2-oleoyl-sn-glycero-3-phosphoethanolamine | POPE |
| | | 1,2-dipalmitoyl-sn-glycero-3-phosphoethanolamine | DPPE |
| | | 1,2-dioleoyl-sn-glycero-3-phosphoethanolamine | DOPE |
| Fatty Acids | FA | (9Z)-Octadec-9-enoic acid | Oleic Acid |
| | | Hexadecanoic acid | Palmitic Acid |
| Enantiomeric Phosphatidylcholine | Ent PC | 3-palmitoyl-2-oleoyl-sn-glycero-1-phosphocholine | Enantiomeric POPC |
| Phosphatidic acid | PA | 1-palmitoyl-2-oleoyl-sn-glycero-3-phosphate | POPA |
| Cardiolipin | CL | 1',3'-bis[1,2-dipalmitoyl-sn-glycero-3-phospho]-glycerol | 16:0 Cardiolipin |
| | | 1',3'-bis[1-palmitoyl-2-oleoyl-sn-glycero-3-phospho]-glycerol | 16:0-18:1 Cardiolipin |
| Sphingomyelin | SM | N/A | N/A |
| Lyso-Phosphatidylcholine | Lyso PC | 1-oleoyl-2-hydroxy-sn-glycero-3-phosphocholine | 18:1-Lyso PC |
| | | 1-palmitoyl-2-hydroxy-sn-glycero-3-phosphocholine | 16:0-Lyso PC |
| Diacylglycerol | DAG | N/A | N/A |

diversity of phospholipid hydrocarbon chain configurations could have impaired growth.

To investigate the importance of headgroup diversity and structure on determining growth rate we provided cells with lipid diets containing diether PG (D.PG) + cholesterol and a mixture of D.PG and D.PC + cholesterol. When lipid extracts of cells grown on D.PG were analyzed by TLC, we observed bands corresponding to both PG and cardiolipin, indicating that cardiolipin synthesis can proceed from D.PG (Fig. 3b). When grown on D.PG and D.PC, TLC analysis showed three bands corresponding to PC, PG, and cardiolipin (Fig. 3c). Thus, cells grown on D.PC generate lipidomes with one phospholipid headgroup, those on D.PG produce lipidomes with two headgroups, and cells grown on both D.PG and D.PC yield lipidomes with three headgroups. Specifically, D.PC cells have only a neutral, bilayer-forming phospholipid, while D.PG cells contain negatively charged phospholipids, including both bilayer-forming (PG) and non-bilayer-forming (cardiolipin) types. In D.PC + D.PG cells, both neutral and negatively charged phospholipids, as well as bilayer- and non-bilayer-forming types, are present. Surprisingly, despite this increased lipidome complexity, the introduction of negatively charged (PG) or conical non-bilayer-forming (cardiolipin) phospholipids did not improve growth rates compared to D.PC cells (Fig. 3a; full growth curves are shown in Supplementary Fig. S4a). To put these values in context, the growth rates we estimate from all of the defined diets considered in this study are more than 10-fold lower than for cells grown on a complex FBS lipid diet, highlighting the effect of reducing lipid diet complexity on growth[6]. Indeed, growth rates for both D.PG and D.PG + D.PC diets were slightly lower than for D.PC alone (Fig. 3a). It is possible that this result is due to the fact that the synthesis of cardiolipin from D.PG results in a single cardiolipin species with hydrocarbon chains of fixed length and saturation, meaning that *M. mycoides* cannot remodel cardiolipin to optimize acyl chain structure[6]. Further, growth of cells on POPC prior to adaptation and ability to synthesize PG and cardiolipin

(Fig. 3d), show only slightly higher growth rates than D.PC diets (Fig. 3a), indicating that reduced growth is not solely due to the introduction of diether phospholipids. Phospholipid headgroup diversity alone, therefore, is not sufficient to rescue growth.

To understand the impact of phospholipid acyl chain diversity on cell growth, we grew cells on a diet of cholesterol and two fatty acids (palmitate and oleate), designated as '2FA'. On this diet, the phospholipidome is predominantly composed of PG and cardiolipin (Fig. 2b), allowing for the synthesis of phospholipids with various acyl chain configurations (e.g., 16:0/18:1 POPG, 16:0/16:0 DPPG, 18:1/18:1 DOPG for PG, and different permutations of PGs as substrates for cardiolipin synthesis). Although DAG and PA are present in low abundances (1.8 mol%, and 0.3 mol%, respectively, Supplementary Data S3), they are also both conical-shaped lipids, similar to cardiolipin. In that respect, their contribution to the membrane's physical properties is relatively small and does not introduce significant differences compared to what is already provided by the far more abundant cardiolipin. Therefore, while the 2FA diet exhibits comparable headgroup diversity to cells grown on D.PG, they exhibit greater acyl chain diversity. Growth rates on the 2FA diet were more than double those on D.PG and approached the growth rates of cells adapted to POPC (Fig. 3a). This suggests that for a lipidome with predominantly two phospholipid headgroups, increased acyl chain diversity can rescue growth. An important consideration in comparing cells grown on the 2FA diet versus the D.PC + D.PG diet is that phospholipids with diether-linked acyl chains are not natural for this organism and could introduce a growth deficit. However, the similar growth rates observed in cells grown on POPC prior to adaptation, which could not yet synthesize PG and cardiolipin, compared to D.PC diets, suggest that reduced growth is not primarily due to the introduction of diether phospholipids. Acknowledging that differences in natural ester linkages and unnatural ether linkages, and the presence of DAG and PA—albeit in low abundance—may also influence the observed growth

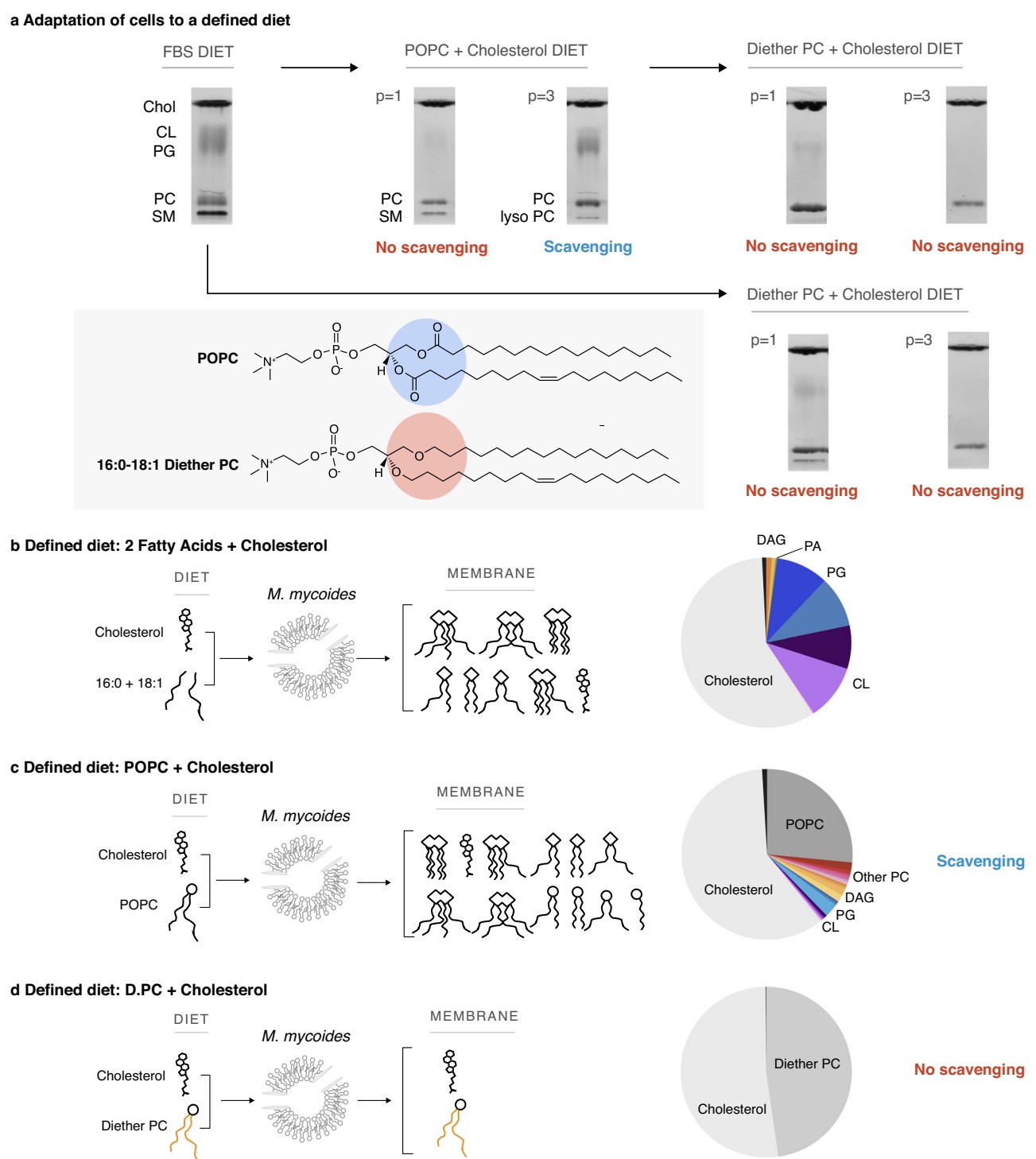

**Fig. 2 | A Defined Lipid Diet Results in a Living Membrane with Two Lipids. a** *M. mycoides* can adapt to defined lipid "diets" with diester or diether phospholipids; resulting in simpler membranes than when grown on a complex diet (e.g. FBS). Adaptation to new diets occurs after 3 passages (*p* > 3). TLC aspect ratio adjusted for legibility; unmodified TLCs can be found in Fig. S2. *M. mycoides* can scavenge acyl chains from POPC, yielding a more complex membrane from a defined diet. **b** Lipidomic analysis of cells grown on only oleic and palmitic acid shows *M.*

*mycoides* cells can synthesize phosphatidylglycerol and cardiolipin species when a source of fatty acids is present. **c** Lipidomic analysis of *M. mycoides* cells grown on a POPC + cholesterol diet shows acyl chain scavenging leads to the synthesis of a diversity of lipids, resulting in a membrane with 28 lipids from a diet of only two. **d** 16:0-18:1 diether PC has ether-linked hydrocarbon chains that cannot be cleaved by *M. mycoides*, eliminating scavenging. Lipidomic analysis shows living cells with two lipids comprising 99.9 mol% of their lipidome.

---

rates, these observations suggest that acyl chain complexity is an important factor in rescuing growth.

Finally, we asked whether growth could also be rescued by providing the full suite of 31 lipids in cells adapted to growth on POPC. It is

possible, for example, that growth is impaired by the disruption of internal phospholipid synthesis or remodeling pathways due to coupling of lipid synthesis with cellular growth. To test this, we took advantage of the fact that when cells grown on the minimal lipid diet

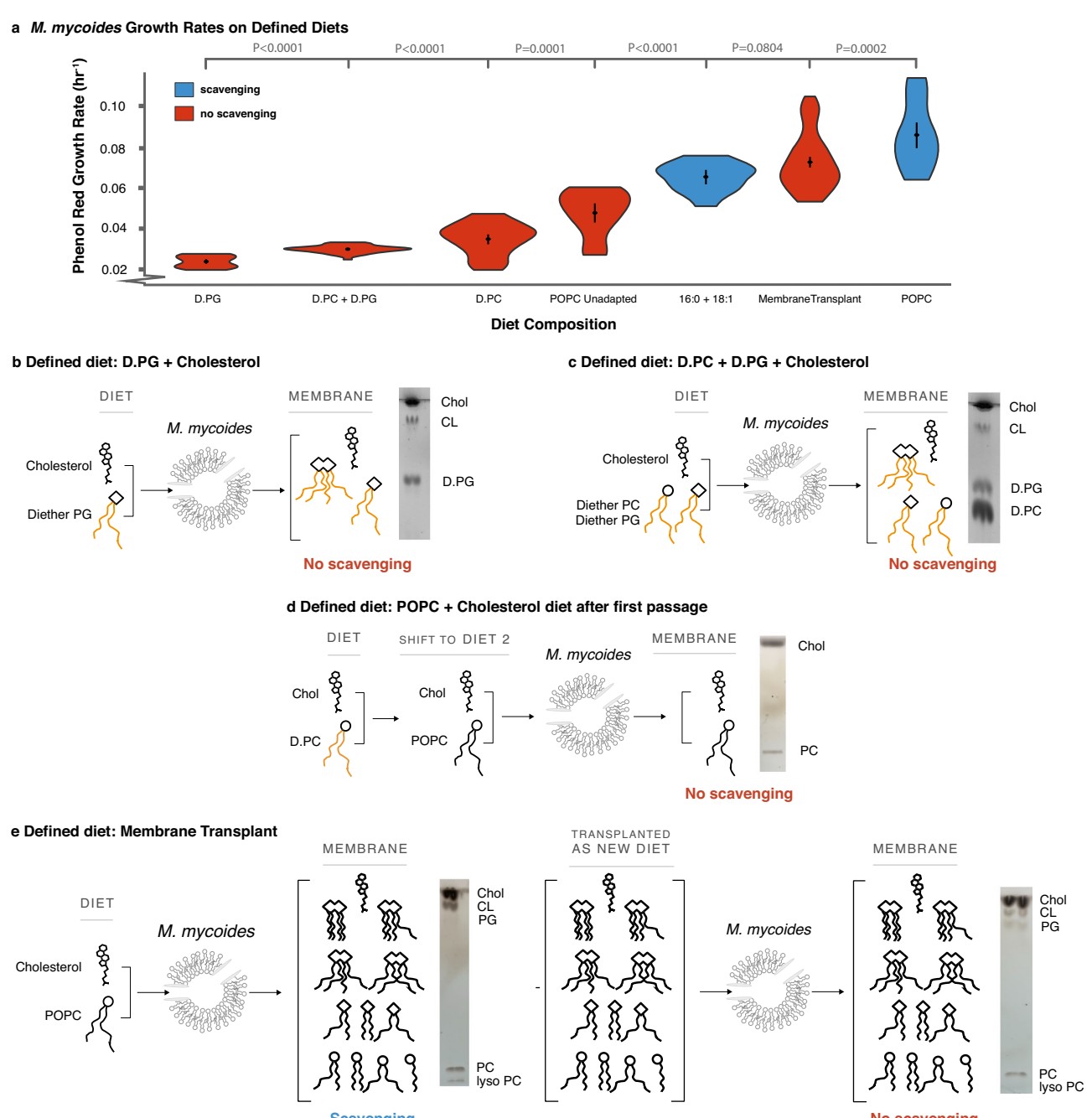

**Fig. 3 | Living membranes with tunable lipid class and acyl chain complexity.**
**a** Reintroducing acyl chain complexity is more effective than headgroup complexity for rescuing growth (relative to cells adapted to a POPC diet), but, even when full complexity is restored, acyl chain scavenging capacity still improves growth. P-values, calculated using a two-tailed Student's t-test, are shown when relevant. *N*, number of replicates, for all growth rates can be found in Supplementary Data S1. **b** 16:0-18:1 Diether PG diet results in the synthesis of cardiolipin with fully ether-linked hydrocarbon chains, increasing head group, but not acyl chain complexity. **c** A diet of both diether PC and PG (D.PC and D.PG) restores the headgroup, but not acyl chain, complexity of the POPC diet. **d** When switched from one diet to another, in this case from the D.PC diet to the POPC diet, there is an adaptation period before acyl chain scavenging occurs. **e** A diet derived from a total lipid extract of cells grown on the POPC diet (a "transplant" diet) restores the full complexity of the POPC. Growth on this transplanted diet was performed after the first passage ($p = 1$) to the new diet, before cells begin to scavenge acyl chains from exogenous phospholipids. Full TLCs can be found in Fig. S2.

are switched to a diet containing ester phospholipids (e.g. POPC + cholesterol) there is a period of adaptation similar to that seen in Fig. 2a. After the first passage, before adaptation, the cells do not yet undergo detectable acyl chain scavenging, and the membrane composition remains predominantly composed of the two lipids POPC and cholesterol (Fig. 3d). This delayed adaptation to acyl chain scavenging allows us a brief window to study simplified membranes composed of lipids *M. mycoides* could normally scavenge acyl chains from. The

simplified membranes only persist until *M. mycoides* is adapted to the new diet after several passages. We thus transferred cells grown on the minimal diet (D.PC + cholesterol) to a lipid diet derived from lipid extracts of cells adapted to growth on POPC (Fig. 3e), and measured growth in the first passage, before cells adapted to scavenge acyl chains from the dietary phospholipids. Growth rates on this transplanted lipidome diet resulted in a nearly complete rescue to levels observed in POPC cells after adaptation (Fig. 3a). These results indicate

that internal phospholipid synthesis and remodeling are not required for optimal growth, and demonstrates a proof-of-principle that functional lipidomes can be engineered and transplanted to living membranes to support growth.

## Minimizing the lipidome of the Minimal Cell

JCVI-Syn3A is a synthetic cell created by the J. Craig Venter Institute (JCVI) by removing every non-essential gene from *M. mycoides*[20]. To do this, Hutchison et al. conceptually divided the genome of *M. mycoides* into 8 segments and systematically went through each, removing genes to see which were essential or non-essential. As such, JCVI-Syn3A is genomically an even simpler model system than *M. mycoides*, while still possessing all of the previously described characteristics of *M. mycoides* that make it a valuable model membrane system[44]. In this study, we used a strain that expresses an additional gene for the fluorescent mCherry protein, to create JCVI-Syn3A-*mCherry*[21,45], which for simplicity we subsequently refer to as JCVI-Syn3A. JCVI-Syn3A provides a good comparison organism to *M. mycoides*, and an experimental platform to examine the role of lipidome composition in supporting the minimal requirements for life. We therefore asked whether the JCVI-Syn3A lipidome could also be minimized.

Since JCVI-Syn3A has a truncated set of genes compared to *M. mycoides*, we first tested whether they retained the capacity to scavenge acyl chains from exogenous phospholipids for the internal synthesis of PG and cardiolipin. Unexpectedly we observed that, like *M. mycoides*, JCVI-Syn3A could still synthesize PG and cardiolipin when fed with a minimal 2-component diet of cholesterol and POPC, as well as with a range of other phospholipid head groups (Supplementary Data S2). This suggests that an undiscovered lipase remains in the JCVI-Syn3A genome. The presence of such lipase activity in a genomically minimal cell could mean that it is essential. Alternatively, it could also indicate that, while not essential itself, it is a secondary "moonlighting" activity of an enzyme with an essential activity[46]. This implies that with the appropriate complement of lipids, the genome could be further minimized by eliminating genes involved in lipid remodeling activities.

Next, we assayed growth of JCVI-Syn3A on a 2-component lipid diet of D.PC + cholesterol. Growth for JCVI-Syn3A and *M. mycoides* on the D.PC + cholesterol diet are commensurate, but on the POPC + cholesterol diet *M. mycoides* has significantly improved growth (Fig. 4a, full growth curves are shown in Supplementary Fig. S4b). By comparison, cells grown on FBS, a complex lipid diet, exhibit ~10-fold higher growth rates[6]. Although growth is exceptionally slow, the culture can be continuously passaged in batch, and samples taken 24 h post-inoculation consistently yield colony-forming units, confirming the viability of the cells (Supplementary Fig S5). Shotgun lipidomic analysis confirmed that JCVI-Syn3A cells grown on a 2-component D.PC + cholesterol diet yielded a lipidome with over 99 mol% composed of only two lipid species (Fig. 4b, Supplementary Data S3). As such, this work has developed the simplest known living membrane in one of the simplest known living organisms.

Previous studies reported that genome minimization in JCVI-Syn3.0 caused pleomorphic traits and abnormalities in cell division, which were rescued by reintroducing 19 genes, resulting in the creation of JCVI-Syn3A[21]. Given that a reduced lipidome could also impair cellular functions, we investigated whether lipidome minimization might lead to abnormal cell morphologies in JCVI-Syn3A. Transmission electron microscopy (TEM) images of JCVI-Syn3A grown on three different lipid diets—fetal bovine serum (FBS), POPC + cholesterol, and D.PC + cholesterol—revealed mostly typical ovoid cells (Fig. 4c, TEM overview images provided in Supplementary Fig. 6a–g). However, two distinct morphological features were observed in subpopulations of cells: internal membrane-encapsulated vesicles and tube-like membranous structures connecting cells.

The tubules, observed in fewer than 20% of cells, were less frequent in cells grown on POPC or D.PC compared to those grown on FBS. These structures are reminiscent of those seen in wall-less L-form bacteria, where they are hypothesized to function as an FtsZ-independent mechanism, possibly representing a primitive form of cell division[47]. Further investigation is required to determine if the tubules observed in JCVI-Syn3A are functional or represent incomplete cell division. Notably, their higher prevalence in FBS-grown cells suggests that they are not directly related to lipidome minimization.

In contrast, the frequency of cells with membrane invaginations increased more than two-fold from ~15% in FBS-grown cells to nearly 40% in those grown on D.PC. In some instances, these invaginations appeared as distinct membrane-encapsulated vesicles within the cell, separate from the cell surface membrane. To confirm the presence of these internalized membrane vesicles, we used cryogenic electron microscopy (cryo-EM) to construct whole-cell tomograms. Tomograms of JCVI-Syn3A from all three diets confirmed the presence of internal membrane-encapsulated vesicles (Fig. 4d, Supplementary Fig. S6h–i). The lower electron density within these vesicles (pixel brightness), compared to the cytoplasm, indicates they result from membrane invagination, encapsulating extracellular fluid.

Interestingly, cells with reduced lipidomes were larger on average compared to those grown on FBS (~0.75 um vs. 0.3 um diameter respectively), as estimated semi-quantitatively from TEM images (Supplementary Fig. S6j). Cells with internal vesicles were particularly enlarged across all conditions (up to 1.5 um average diameter). These results suggest that lipidome minimization leads to larger cell sizes and a higher frequency of membrane invaginations, indicative of impaired regulation of cell size and shape. The increased frequency of membrane invaginations could result from non-optimal membrane bending rigidity or intrinsic curvature from the loss of cardiolipin, and acyl chain diversity. Nonetheless, the fact that around half of the observed cells maintained normal morphology shows that even with just two lipid species, JCVI-Syn3A is capable of preserving typical cell morphology.

## Tuning lipid chirality

Having explored the minimal requirements for lipidome complexity in *M. mycoides* and JCVI-Syn3A, we next sought to leverage these model systems to probe another fundamental aspect of lipid biology: chirality. Glycerolipids (including phospholipids) have a chiral center in the glycerol backbone leading to enantiomeric lipids that are mirror images of each other (Fig. 5a). In bacteria and eukaryotes, glycerolipids are synthesized with acyl chains at the sn-1 and sn-2 positions, and the phosphate head group at the sn-3 position (Fig. 5a: G3P enantiomer). Conversely, archaea synthesize glycerolipids with the phosphate at the sn-1 position and the acyl chains at the sn-2 and sn-3 positions (G1P enantiomer)[48,49]. Known as the 'Lipid Divide', this difference in stereochemistry between lipids of archaea and the rest of life has long stood as an unexplained enigma[50–53]. Did a last common ancestor have membranes with both enantiomers? What are the consequences of having a racemic mixture of phospholipids in a living membrane? To date, no naturally occurring organism has been found that has comparable amounts of both enantiomers in its membrane. Furthermore, experimentally modulating phospholipid chirality in the lipidome through genetic approaches presents significant challenges, as shown by the work of Caforio et al.[53]. However, the ability of mycoplasma to uptake exogenous lipids makes them an exceptionally well-suited model for unraveling this elusive problem in membrane biology.

To establish whether *M. mycoides* and JCVI-Syn3A can grow on a G1P phospholipid enantiomer, we prepared a lipid diet consisting of cholesterol and enantiomeric POPC (entPOPC), as well as a racemic mixture of cholesterol and POPC:entPOPC (1:1 mol%). We compared the two diets containing entPOPC against growth on cholesterol + POPC (Figs. 5b, 5c). Growth rates derived from Phenol Red absorbance of both organisms show that the introduction of entPOPC to the lipid diet results in impaired growth, with the racemic diet yielding the most

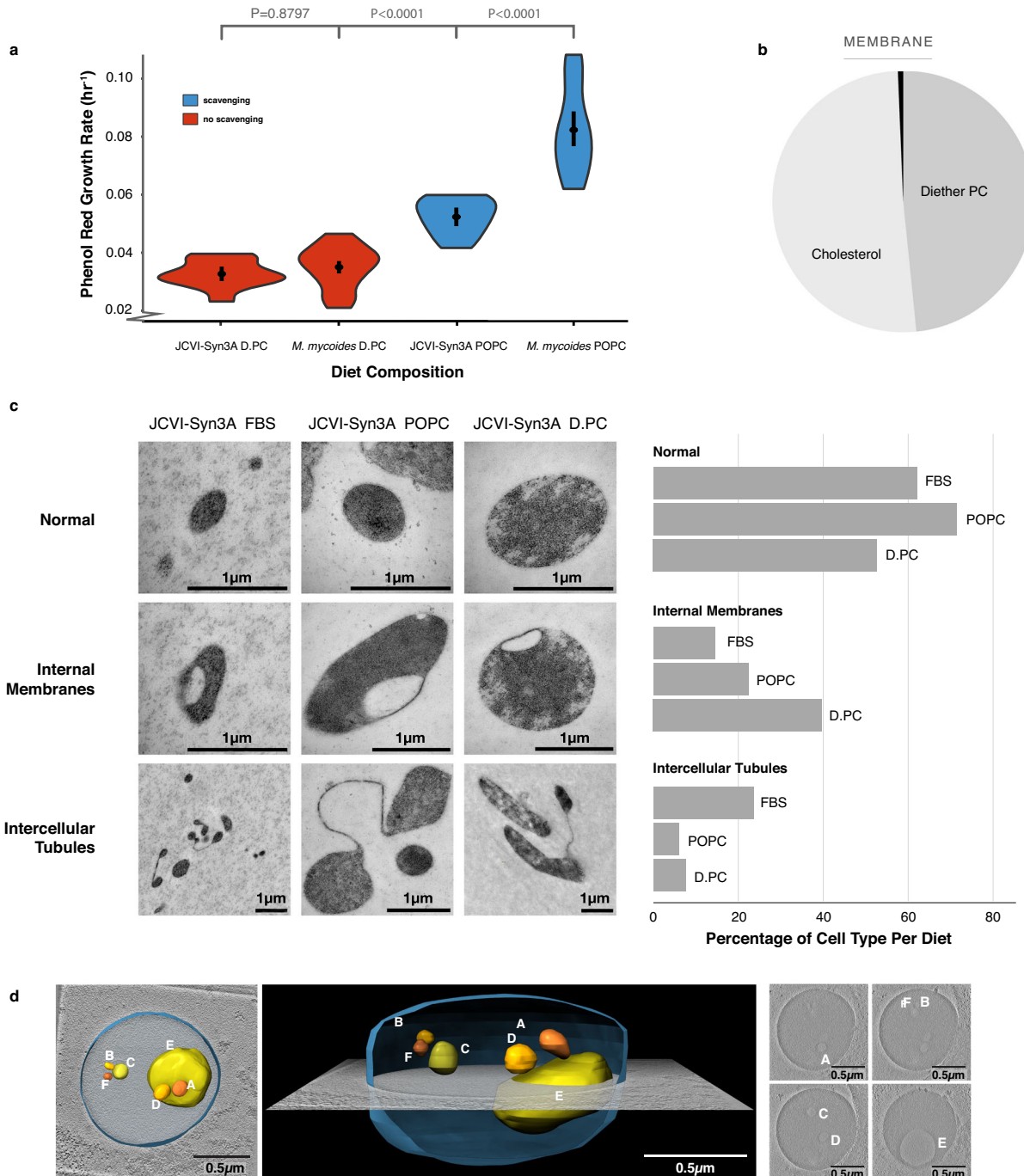

**Fig. 4 | Lipidome Minimization in JCVI-Syn3A: Achieving a Minimal Membrane in a Genomically Minimal Cell. a** JCVI-Syn3A exhibits growth rates similar to *M. mycoides* on a minimal lipid diet. **b** JCVI-Syn3A is viable when grown on a diether-PC (D.PC) + cholesterol diet with 99.34 mol% of the membrane comprised of only two lipids. **c** JCVI-Syn3A cells display three phenotypes when observed with Transmission Electron Microscopy; a normal ovoid morphology, cells with internal membranes, and cells with intercellular tubules. Cell counts reveal the D.PC + cholesterol diet yields significantly more cells with internal membranes than the other diets. Cell counts can be found in Fig. S6a–g **d** A tomogram of a JCVI-Syn3A cell on the D.-PC + cholesterol diet with internal membranes reveals a large cell with multiple membrane-bound internal vesicle-like structures that have a lower electron density compared to the rest of the cell, suggesting the internal membrane-bound structures are vesicles occurring from membrane invagination. Modeling shows these vesicles are completely enclosed and separate from the cell surface membrane. Images in this figure are from a single biological replicate. *P*-values, calculated using a two-tailed Student's t-test, are shown when relevant. *N*, number of replicates for all growth rates, can be found in Supplementary Data S1.

pronounced decrease in growth rate. We also evaluated growth by measuring optical density, as a proxy for cell density (Supplementary Fig. S7), which confirmed a reduction in growth with the introduction of entPOPC. However, in contrast, growth was similar for the entPOPC and racemic lipid diets. This difference likely reflects the fact that Phenol Red growth rate estimates reflect metabolic activity, which is not necessarily coupled with the production of cell biomass measured by optical density. Thus, we demonstrate that cells are viable when fed enantiomeric phospholipids.

We next asked how the introduction of entPOPC affected the mechanical robustness and permeability of the membrane. To assay membrane robustness, we measured sensitivity of cells to

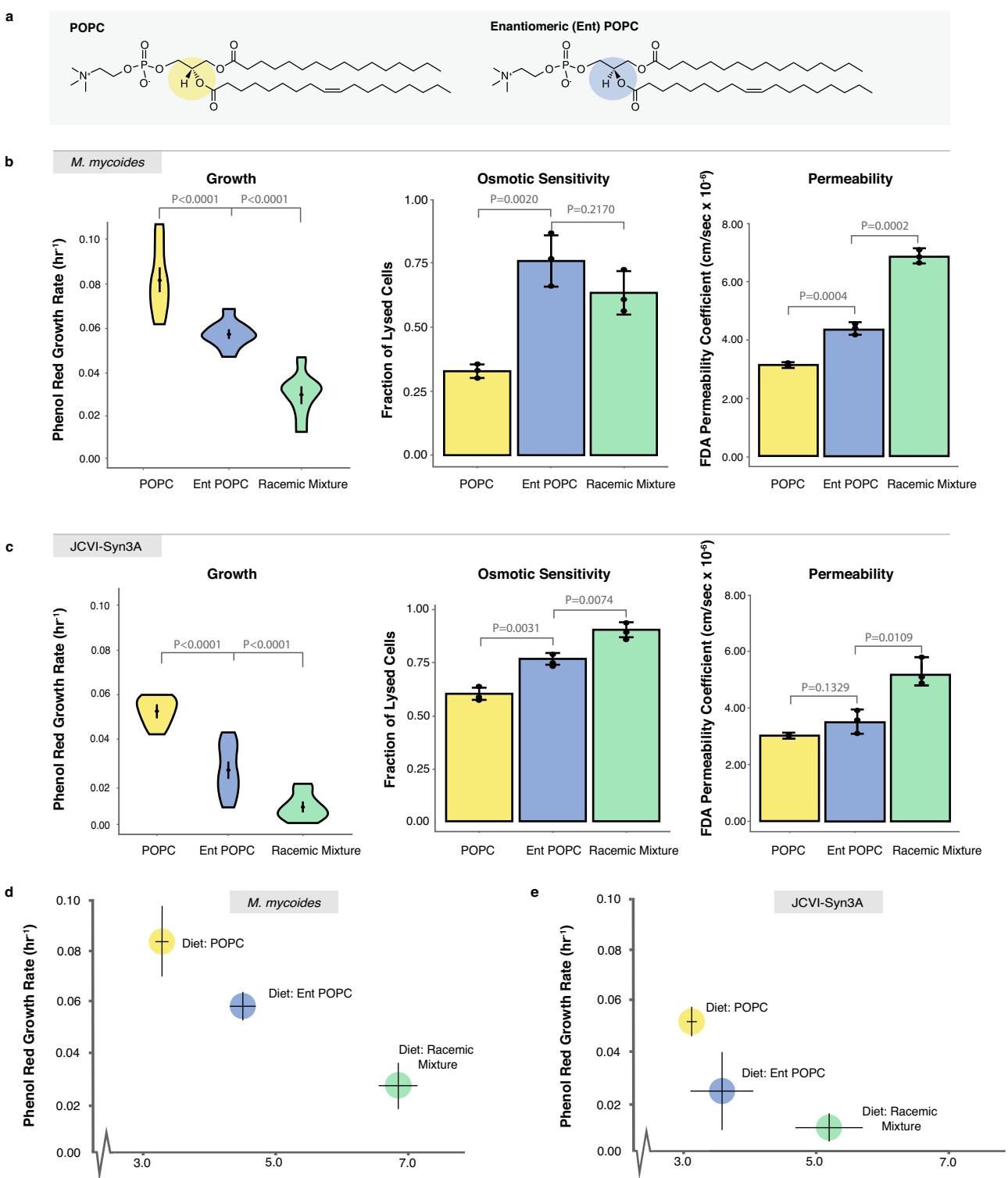

**Fig. 5 | Enantiomeric Lipid Diets Negatively Affect Cell Growth and Membrane Properties. a** Enantiomeric (ent) POPC is a synthetic chiral POPC with the head group in the sn-1 position, rather than sn-3 position. **b** *M. mycoides* and **c** JCVI-Syn3A cells exhibit slower growth when grown on diets with entPOPC present; are more fragile to hypoosmotic shock when grown on diets with entPOPC present; and are more permeable to non-chiral fluorescein diacetate (FDA) when grown on diets with entPOPC present. **d** *M. mycoides* and **e** JCVI-Syn3A cell growth and membrane permeability are inversely correlated. P-values, calculated using a two-tailed Student's t-test, are shown when relevant. N, number of replicates for all growth rates, can be found in Supplementary Data S1. N = 3 biological replicates for osmotic sensitivity and permeability subfigures.

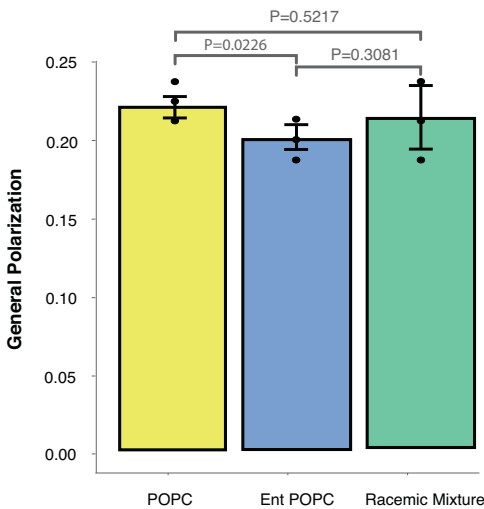

**Fig. 6 | Enantiomeric lipids have no significant effect on lipid order.** Lipid vesicles reconstituted from total lipid extracts of *M. mycoides* cells had a similar lipid order (measured as the general polarization index of c-laurdan) for all three diets with POPC enantiomers. *P*-values, calculated using a two-tailed Student's t-test, are shown. $N = 3$ biological replicates (membranes extracted from 3 biological *M. mycoides* replicates).

hypoosmotic shock, by determining what fraction of cells were lysed following a shock. Since *M. mycoides* and JCVI-Syn3A lack a cell wall or cytoskeleton, cell lysis during rapid hypoosmotic shock is indicative of membrane rupture strength. Furthermore, mechanosensitive ion channels which can protect cells from hypoosmotic shock have, to the best of our knowledge, not been reported in either organism, or annotated in the genomes, and are not present in all *Mycoplasmas*[46,54]. Therefore, it is reasonable to cautiously interpret the susceptibility to lysis from hypoosmotic shock as indicative of membrane stability. However, we cannot rule out the possibility that there are undiscovered mechanosensitive channels, and this would suggest that mechanosensitive gating is sensitive to lipid chirality. Hypoosmotic sensitivity increased significantly for cells grown on entPOPC diets, indicating that the enantiomeric lipids affect the mechanical robustness of the membrane and, consequently, the whole cell (Fig. 5b,c). To assay permeability, we measured the rate of permeation of fluorescein diacetate (FDA), a non-chiral molecule, across the cell membrane[55,56]. Membrane permeability, as measured by the permeability coefficient of FDA increased significantly upon the introduction of enantiomeric lipids, but was the highest for both *M. mycoides* and JCVI-Syn3A for the racemic lipid diet (Fig. 5b,c). When plotted against growth rate, there is an apparent correlation of lower growth with higher permeability (Fig. 5d,e), possibly indicating membrane leakiness as one of the factors underlying impaired growth. The permeability coefficients were similar to those previously reported in mammalian cells[56].

Increased permeability and reduced membrane robustness could be due either to changes in the properties of the lipid bilayer, or through changes in lipid-protein interactions. Previous work on PC enantiomers in model membranes revealed that modest changes in permeability to calcein occur in membranes composed of both enantiomers, in particular scalemic (not 1:1) mixtures. However, such small changes would not account for the large change in permeability we observed. To determine if introducing enantiomeric POPC affected membrane bilayer robustness or permeability we reconstituted cellular lipid extracts from *M. mycoides* into liposomes and evaluated c-laurdan fluorescence (Fig. 6). The c-laurdan General Polarization index (GP) reports bilayer hydration, which is closely coupled with permeability and mechanical robustness of the membrane[57,58]. Surprisingly, GP values did not vary significantly across all three POPC

diets, indicating that the lipid bilayer itself was not disrupted by changes in chirality, and implicating an effect on lipid-protein interactions. Indeed, several studies have demonstrated an effect of varying lipid chirality on lipid-peptide interactions, and the permeability of membranes to chiral amino acids[59,60]. Thus, disrupted lipid-protein interactions are the most likely basis for the observed phenotypes, setting the stage for future work employing *M. mycoides* and JCVI-Syn3A as model systems to explore the significance of lipid chirality on lipid-protein interactions in a living membrane.

## Discussion

In this study, we introduce *M. mycoides* and the Minimal Cell JCVI-Syn3A as simple model organisms with tunable lipidomes for studying the role of lipid complexity. By choosing a model membrane system incapable of synthesizing or modifying fatty acids, and developing a set of defined lipid diets for that system, we have demonstrated the creation of a platform in which the lipidome can be tuned in terms of phospholipid headgroup, acyl chain composition or lipid chirality. Using this platform, we created the simplest living membrane, and one incapable of undergoing acyl chain remodeling. These tunable living membranes allowed us to quantitatively examine the contribution of lipidomic features to the fitness of a minimal living system.

Our observations broadly demonstrate how lipidome composition is crucial even for relatively simple microorganisms. This is anecdotally illustrated by the capacity we observe for mycoplasma to produce a complex lipidome from a single exogenous phospholipid, which is an impressive evolutionary adaptation to their pathogenic lifestyle. Indeed, this acyl chain scavenging activity, which confounded our initial attempt to simplify the lipidome, suggests a potential target for treating mycoplasma infection. Although mycoplasma can survive in pure culture with a reduced lipidome, they may fare much worse in the context of a host immune system. Furthermore, the ability for the Minimal Cell to grow with a lipidome of two lipids implies the possibility for further minimization of the genome, through the deletion of pathways involved in the scavenging of phospholipid acyl chains and the internal synthesis of phospholipids.

A long-standing challenge in membrane biology has been to understand why life has evolved such complex lipidomes and to identify the essential features of lipidomes required for optimal membrane function and cellular fitness. The simplest lipidome so far reported was from a Gram-negative bacterium, composed of 27 lipid species, excluding outer membrane lipopolysaccharides that were not analyzed[1]. By reducing the lipidome of *M. mycoides* and JCVI-Syn3A down to two lipids, we show that a complex lipidome is not essential for life, but that two lipids are far from optimal.

Through systematic reintroduction of lipidomic features into cells with a minimal lipidome, we demonstrated that phospholipid headgroup diversity alone does not significantly rescue growth. Consistently, minimizing headgroup diversity in *B. subtilis* has little effect on growth[13]. Similarly, *E. coli* mutants lacking phosphatidylglycerol (PG) and cardiolipin can be viable and often do not exhibit significant growth deficiencies[7,61]. It was perhaps most surprising that *M. mycoides* grew comparably well in the absence of zwitterionic lipids (PC), which are essential for *E. coli* to support proper membrane insertion and activity of membrane proteins[62]. These observations emphasize the limited impact of headgroup diversity, at least in simple bacterial organisms, in the absence of other critical lipid features.

Restoring acyl chain diversity, in contrast, did enhance growth rates. This finding is consistent with recent work showing the importance of acyl chain unsaturation in various cellular processes, including the assembly and function of the nuclear pore complex in eukaryotes[63] and in neuronal membranes[64]. The enhanced growth observed when acyl chain diversity was restored in *M. mycoides* suggests that while headgroup diversity alone is insufficient, it may contribute to improved cellular fitness when coupled with the appropriate acyl chain

composition. For example, the introduction of cardiolipin or PG could potentially improve growth, but only if these lipids are present with the correct acyl chain configurations that support optimal membrane properties and lipid-protein interactions. The approach to tuning the lipidome that we introduce here provides a platform to explore the synergistic effects of specific acyl chain-headgroup combinations, to unravel how these lipidomic features can be optimized for membrane function and cellular fitness in minimal systems.

By applying a targeted chemical approach to reduce the lipidome of JCVI-Syn3A, we demonstrated the feasibility of further simplifying the molecular composition of a genomically minimized organism. The observation that a minimal cell membrane can function with only cholesterol and one species of PC demonstrates that the fundamental requirements for life can be achieved with a remarkably simple lipid composition. For synthetic biology, this insight simplifies the challenge of designing synthetic cells, revealing the potential to create functional living membranes with minimal components. This work lays a foundation for future efforts to understand how minimal lipidomes can be optimized in synthetic and engineered biological systems

One limitation of this work is that the growth medium is not defined, and there is a very small contribution of lipids from components of the growth medium, such as the yeast extract. Thus, while the majority (>99%) of the minimal lipidome is composed of 2 lipids, there is a fraction of a percent of very low abundance lipids derived from the media and it is possible, but unlikely, that these trace lipids play a significant role. In this regard, development of a defined growth medium will be essential in the continued development of mycoplasmas as minimal model membrane systems[33]. Another factor limiting the minimization of the lipidome to one lipid is that cholesterol (or an analog) is required for growth, but also cannot form a bilayer alone. So, it is possible that even one bilayer-forming lipid could support life in a cell that has not evolved to require sterols. It is also, however, possible that having a membrane-reinforcing sterol or sterol analog is critical for building a stable cell membrane with only one phospholipid. A mycoplasma-like organism such as *Mesoplasma* that does not require sterols for growth would provide a means to test this hypothesis[65–67]. Nonetheless, a JCVI-Syn3A membrane with two lipids comprising over 99% of the lipidome is currently the simplest living membrane that has been reported.

The lipid divide represents another major problem in membrane biology. Homochirality is a fundamental feature of biomolecular chemistry. Biomolecules exhibit enantioselectivity for chirally compatible interaction partners. Consequently, life has evolved to rely on homochiral molecules (e.g., L-amino acids and nucleic acids). An interesting twist is in the divide between the chirality of phospholipids made by Archaea and the rest of life[68]. The divide raises questions about whether the last universal common ancestor (LUCA) possessed a heterochiral lipidome, or whether the divide happened after the divergence of Bacteria and Archaea[69,70]. Further, current theories that eukaryotes emerged from an archaeal ancestor[71], create a conundrum in explaining why Eukaryotes don't have archaeal lipid enantiomers, and whether a gradual transition involving heterochiral lipidomes could have occurred. At the crux of these conundrums is whether heterochiral lipidomes can support stable membranes and optimal cellular fitness.

We took advantage of the tuneability of *M. mycoides* and JCVI-Syn3A lipidomes to observe how heterochiral lipidomes influence the membrane and cellular growth. Previous work in living systems has achieved lipidomes with a mixture of structurally diverse bacterial and archaeal lipids that differed not only by chirality, but also in a variety of other ways[53]. A unique feature of this study is that we were able to introduce two enantiomers of a single lipid structure (POPC) into a living membrane. Our results demonstrate that a heterochiral lipidome results in a leakier membrane and impaired cellular robustness and growth. In contrast, previous work in *E. coli* indicated that heterochiral

lipidomes did not affect growth[53]. *E. coli* has an outer membrane and cell wall that might compensate for lack of stability caused by heterochirality. The lipids in the *E. coli* study were not stereoisomers, but rather entirely different lipid structures (ether vs. ester acyl chain linkages, isoprenoid tails vs. fatty acid tails). It is possible that these structural differences somehow obscured the destabilizing effects of heterochirality. Ultimately, the basis for these diverging observations will provide insight into the biological significance of lipid chirality. Nonetheless, our work shows that in a minimal organism with a single membrane, heterochiral lipidomes can support growth, but lead to impaired robustness and fitness. Interestingly, this suggests that LUCA could have existed before the evolution of homochiral cellular membranes. This possibility would eliminate the need to view the lipid divide between bacteria and archaea as resulting from independent evolutionary events. Instead, it suggests the ancestral lipidome could have been heterochiral, consistent with a simpler path for the evolution of modern membranes. Additionally, the reduced fitness and increased membrane permeability resulting from heterochiral lipidomes suggests selective pressure against such membranes in ancestral organisms, favoring the evolution of the homochiral membranes characteristic of modern life.

The changes in membrane robustness and permeability that we observed do not seem to be explained by changes in lipid order of pure lipid vesicles reconstituted from cell lipid extracts of homo- and heterochiral lipidomes. This implies that lipid-protein interactions are enantioselective and may be affected by lipid chirality in ways that impair membrane robustness and function. Numerous in vitro studies have demonstrated the effect of lipid chirality on lipid-protein interactions, consistent with the possibility that the phenotypes we observed are rooted in perturbed lipid-protein interactions. Looking forward, *M. mycoides* and JCVI-Syn3A will be excellent model systems to explore the functional consequences of lipid chirality on lipid-protein interactions.

Our approach to employ *M. mycoides* and JCVI-Syn3A as minimal living model membrane systems paves a new path towards unraveling the role of lipidome diversity and complexity. Our observations reveal that life does not require complex lipidomes. However, minimization comes with clear trade-offs in cellular fitness. We further demonstrate the capacity of these model systems to serve as chassis for exploring fundamental questions in membrane biology. The ability of this system as a testing platform for lipid diets with a variety of features (including class and acyl chain composition) makes it a useful and simple in vivo model for design-test-build applications of membrane composition experiments. Furthermore, we demonstrated the ability to force *M. mycoides* and JCVI-Syn3A to take up and incorporate enantiomeric lipids in their membranes, the first time this has been shown in eukaryotic or prokaryotic organisms, and an exciting first step to allow us to probe questions about the lipid divide and the role of chirality in membrane stability and cell fitness. Overall, we have demonstrated the creation of a simple, tunable, living, model membrane system that can be used as a novel platform for probing the design principles of living membranes.

## Methods

### Cell culture
*M. mycoides* and JCVI-Syn3A were grown in liquid culture at 37 °C on a modified SP4 growth medium with Phenol Red[72]. The lipid source was provided either by Fetal Bovine Serum (FBS), a complex natural source of mammalian lipids, or by complexing lipids with delipidated bovine serum albumin (BSA). Lipids were dissolved in ethanol and added to the medium at 37 °C immediately before the addition of cells, in the concentrations given in Supplementary Data S1. *M. mycoides* was passaged to new growth medium by adding 50 μL of cell culture at OD600 0.4–0.8 into 7 mL of freshly prepared media in a T25 Flask (Stand., Vent. Cap) or 10 mL of media in a Duran 100 mL glass flask.

This corresponded to roughly one passage per day. JCVI-Syn3A was passaged to new medium by adding 200 μL of cell culture at OD600 0.2–0.6 into 7 mL of freshly prepared media. This corresponded to one passage every two to four days, depending on cell growth rates. Cells were grown at 37 °C and 40 RPM in a Kuhner shaker incubator.

## Growth rates

**Metabolic growth index.** Due to differences in cell size and behavior between *M. mycoides* and JCVI-Syn3A, measuring growth curves with optical density by taking the absorbance at 600 nm ($OD_{600}$) proved to be an unreliable method to achieve growth estimates that were robustly comparable between the two organisms. Phenol Red is a pH indicator that is commonly used as a readout for growth in mycoplasma cell culture[73,74] as it detects changes in pH as metabolic activity acidifies the culture media through absorbance at 562 nm ($A_{562}$). To obtain the rate of pH change ($A_{562}$) over time cells were grown on the liquid handler system Biomek i7 Automated Workstation in 96-well plates (square well, clear bottom, 400 uL media volume, 100 rpm shaking), and the absorbance was recorded hourly to record changes in phenol red absorbance as cell growth acidifies the media (Supplementary Fig. S3a)[19,72,73]. All absorbances were then plotted with respect to time, and then fitted to the logistic function $N_t = K (1 + (K \cdot N_0)/N_0) e^{-rt}$ where $N_t$ is population at time t, K is the carrying capacity, $N_0$ is the initial population size, and r is the growth rate; or, in this case, the rate of pH change. Only the exponential part of the curve was considered to exclude lag and stationary phase. To validate the relevance of this method as a readout for cell growth, growth rates for a representative set of *M. mycoides* diets were calculated using $OD_{600}$ and were compared to the rates calculated with Phenol Red absorbance. The result was a strong linear correlation between the two estimates (Supplementary Fig. S3b). Representative curves of the entire period of growth for *M. mycoides* and JCVI-Syn3A on three different diets and generated by the two methods is also shown (Supplementary Fig S5). Cell growth data shown in Supplementary Fig. S5, measured by Phenol Red $A_{562}$ and $OD_{600}$ were obtained through hourly manual measurements from cultures grown in 30 mL in a Duran 100 mL glass flask in a Kuhner shaker incubator at 37 °C and 40 rpm shaking using a DeNovix Spectrophotometer.

## Lipid extraction and thin layer chromatography

Lipids were extracted using a Bligh-Dyer lipid extraction protocol[75]. When appropriate, lipid concentration was determined with a phosphate assay. TLC plates with Silica Gel and a 10x concentrating zone were pre-washed with chloroform to remove debris and other artifacts, and were then dried at 60 °C for > 30 min to ensure complete solvent evaporation. Lipid samples were then loaded on the plate and were dried at 60 °C for > 10 min to again ensure complete solvent evaporation. The plate was then run in a closed glass chamber using a running solution of chloroform:methanol:acetic acid:water (85:25:5:4), and was subsequently dried again at 60 °C for > 30 min. TLC plates were then rinsed with a 3% copper acetate and 8% phosphoric acid solution and heated by a handheld heat gun at 280 °C to char lipid spots.

**Liposome preparation.** Lipids of a known concentration (either in stock solutions or extracted from cells and validated with a phosphate assay) were prepared in a 2 mL glass by having their solvents evaporated overnight under a 10–17 mbar vacuum. Lipids were then resuspended in the appropriate volume of Liposome buffer (LiB) (10 mM HEPES, 100 mM NaCl) to achieve 100 mM liposome solution and incubated for 30 min at 37 °C. Subsequently, liposomes were homogenized with ten freeze-thaw cycles (1 min in liquid nitrogen followed by 5 min at 60 °C) and 7 extrusion cycles in a Hamilton syringe setup through a 100 nm filter.

**Propidium iodide osmotic shock assay.** 3 × 0.4 ODU of cells with OD 0.2–0.4 (1 ODU = 1 mL of cells at OD 1) were harvested and spun down in pre-warmed centrifuge at 5000 g, 7 min, 37 °C, with slow acceleration and slow deceleration. 1 μL of 1 mM Propidium Iodide dye was added to appropriate wells on a black or clear bottom 96-well plate. The supernatant was aspirated and each cell pellet was resuspended in 400 μL of either H2O or 5X diluted mycoplasma wash buffer (MWB) (original MWB: 20 mM HEPES, 200 mM NaCl, 1% W/V Glucose). An extra tube with cells resuspended in 400 μL H2O and boiled at 95 °C in a thermoshaker at 1000 rpm for 10 min was set up as a positive control. During that time, all other tubes were incubated on the 37 °C thermoshaker at 600 rpm. 100 μL of the cells in the appropriate buffer was added to each of their three analytical wells on the 96-well plate. Cells were incubated for 30 min with shaking at 37 °C in a TECAN Spark 20 M plate reader and the fluorescent signal was subsequently measured at excitation 539 nm emission 619 nm.

**Fluorescein diacetate permeability assay.** 3 × 0.4 ODU of cells with OD 0.2–0.4 were harvested and spun down in pre-warmed centrifuge at 5000 g, 7 min, 37 °C, with slow acceleration and slow deceleration. Cells were washed 1× in MWB and resuspended in 400 μL MWB. A Fluorescein standard curve was prepared on the plate in the following concentrations: 0, 0.25, 0.5, 1.2, 2 μM Fluorescein. 100 μL resuspended cells were added to the proper wells (in triplicate), and 100 μL MWB was added to the Fluorescein standard wells. FDA was added to cell wells to achieve a final FDA concentration of 5 μM. Cells were incubated on the TECAN Spark 20 M at 37 °C for 140 min and measurements were taken every 20 min at excitation at 485 nm and emission at 525 nm. The permeability coefficient (P) of FDA was calculated using Fick's Law ($Q = P*A*(C_{out} - C_{in})$) where Q is the flux across the membrane, and is a constant as the slope of fluorescein increase over time was linear; A is the area of the membrane, calculated using the assumption that the cross sectional area of POPC + cholesterol in a bilayer is 45.1 $Å^{76}$, the ratio of POPC: cholesterol is roughly 1:1, the area of POPC + cholesterol is equivalent to the area of Enantiomeric POPC + cholesterol, and the number of POPC molecules can be calculated from the phosphate assay described below; $C_{out}$ and $C_{in}$ are the concentrations of FDA outside and inside the cell (respectively); and $Ci_n$ is 0 as FDA is immediately converted to fluorescein upon entering the cell.

**C-laurdan general polarization assay.** Liposomes from whole cell lipid extracts were incubated at 37 °C and 1000 rpm for 10 min on a tabletop thermoshaker. While cells were incubating, c-laurdan was removed from a −20 °C freezer and warmed to room temperature on the bench. 1 mM c-Laurdan stock in EtOH was diluted 4× and 1 uL of diluted stock for a final molarity 0.5 μM c-laurdan was added to 500 μL of liposomes. Samples were incubated at 37 °C and 1000 rpm shaking for 10 min on a tabletop thermoshaker. 100 μL of each sample was added to a flat bottom 96-well plate well with analytical triplicates for each biological replicate. The signal was measured on a TECAN Spark 20 M with a 2-channel fluorescence reading with excitation at 385 nm and emission at 440 and 490 nm respectively. General polarization (GP) was calculated using the formula: $GP = (I440 − I490)/(I440 + I490)$ where *I* is the fluorescence emission intensity at the respective wavelength.

**Phosphate assay.** To estimate the concentration of phospholipids in a certain amount of cells, the amount of phosphate in the lipid extraction of 1 OD Unit (ODU: 1 mL of $OD_{600} = 1$) of cells was measured. To measure phosphate amount a modified version of the method of Chen et al. was used[77]. The lipid extraction was added to Pyrex glass tubes with both biological and analytical triplicates (3 vials per biological replicate), and the solvent was evaporated by a brief (<5 min) incubation at 200 °C. 50 μL of water was added to each sample. To prepare a standard curve of phosphate amounts, an ICP phosphorus standard

was diluted in water to give 5, 10, 20, 50, 100, and 200 nmols of phosphate. 500 μL of 70% perchloric acid was added to each vial and, after brief vortexing, samples were incubated at 200 °C for 120 min. Tubes were cooled down in ice water, and 1 mL of 10% w/v ascorbic acid followed immediately by 1 mL of 2.5% w/v ammonium heptamolybdate were added (with a brief vortex after adding each reagent). Samples were incubated at 37 °C for 30 min, and then absorbance at 820 nm was measured on a TECAN Spark 20 M plate reader by adding 200 μL of each tube to a clear-bottom 96-well plate. A standard curve of absorbance at 820 vs phosphate amount was calculated using the phosphate standard, and lipid amount of each sample was calculated based on the absorbance value, standard curve, and assumption that one phosphate molecule equals one phospholipid.

**Lipidomic analysis.** Lipids extracted from cells using the aforementioned Bligh-Dyer protocol were submitted to Lipotype for mass spectrometry-based analysis[78]. The general procedure is described in Sampaio et al. 2011[2]. For the analysis, samples were spiked with internal lipid standard mixture containing: cardiolipin 14:0/14:0/14:0/14:0, ceramide 18:1;2/17:0, diacylglycerol 17:0/17:0, hexosylceramide 18:1;2/12:0, lyso-phosphatidate 17:0, lyso-phosphatidylcholine 12:0, lyso-phosphatidylethanolamine 17:1, lyso-phosphatidylglycerol 17:1, lyso-phosphatidylinositol 17:1, lyso-phosphatidylserine 17:1, phosphatidate 17:0/17:0, phosphatidylcholine 17:0/17:0, phosphatidylethanolamine 17:0/17:0, phosphatidylglycerol 17:0/17:0, phosphatidylinositol 16:0/16:0, phosphatidylserine 17:0/17:0, cholesterol ester 20:0, sphingomyelin 18:1;2/12:0;0, triacylglycerol 17:0/17:0/17:0, and cholesterol D6. After extraction, the organic phase was transferred to an infusion plate and dried in a speed vacuum concentrator. It was then resuspended in 7.5 mM ammonium acetate in chloroform/methanol/propanol (1:2:4, V:V:V) and a 33% ethanol solution of methylamine in chloroform/methanol (0.003:5:1; V:V:V). All liquid handling steps were performed using Hamilton Robotics STARlet robotic platform with the Anti Droplet Control feature for organic solvents pipetting. Samples were analyzed by direct infusion on a QExactive mass spectrometer (Thermo Scientific) equipped with a TriVersa NanoMate ion source (Advion Biosciences). Samples were analyzed in both positive and negative ion modes with a resolution of $Rm/z = 200 = 280000$ for MS and $Rm/z = 200 = 17500$ for MSMS experiments, in a single acquisition. MSMS was triggered by an inclusion list encompassing corresponding MS mass ranges scanned in 1 Da increments. Both MS and MSMS data were combined to monitor CE, DAG and TAG ions as ammonium adducts; PC, PC O-, and Diether PC as acetate adducts; and CL, PA, PE, PE O-, PG, PI and PS as deprotonated anions. Diether PC was quantified using the diester PC standard mentioned above. Although both lipids share the same headgroup chemistry, differences in ester and ether linkages may cause slight variations in ionization efficiency and mass spectrometric response. As a response factor calibration was not performed, the quantification of diether PC should be regarded as semi-quantitative. MS only was used to monitor LPA, LPE, LPE O-, LPI and LPS as deprotonated anions; ceramide, hexosylceramide, sphingomyelin, LPC and LPC O- as acetate adducts and cholesterol as an ammonium adduct of an acetylated derivative[79].

**Room temperature TEM**
JCVI-Syn3A cells were adapted to FBS, POPC + cholesterol, and D.PC + cholesterol diets and harvested at between $OD_{600}$ 0.05–0.2 at volumes to achieve 1ODU of cells for each diet. Cells were spun down and resuspended in 1 mL Mycoplasma Wash Buffer without glucose (20 mM HEPES, 200 mM NaCl). Cells were pre-fixed 30 min at room temperature with a final concentration of 0.5% glutaraldehyde (25% aqueous stock solution (EMS), directly added to the culture medium). Then cells were spun down for 7 min @ 7000 g @ 37 °C in a centrifuge (Heraeus Biofuge PrimoR with swing-out buckets). The cell pellet was sucked into cellulose capillary tubes with an inner diameter of 200 μm

and a permeability cut-off >5 KD (Leica). Filled capillaries were chopped into small pieces with a scalpel blade, at the same time sealing the capillary ends. Capillary pieces were rapidly frozen using Leica EM-ICE high pressure freezing device in 6 mm–aluminum carriers with 200 μm deep depression (Leica) using hexadecene as a filler. Freeze substitution was done in an AFS 2 freeze substitution device (Leica) in 1% osmium tetroxide in acetone starting at −90 °C and raising the temperature to 0 °C over 72 h. Samples were infiltrated gradually with 25%, 50%, 75%, 100% EMbed 812 (Science Services) in acetone over two days. Capillaries were infiltrated with pure resin over two days and embedded in double-end silicon molds (TED PELLA INC) and polymerized at 60 °C for 48 h. 70 nm–sections were cut with a Leica EM UC6 and mounted on formvar-coated copper slot grids (EMS). Sections on the grids were contrasted with uranyl acetate and lead citrate prior to imaging. Imaging was done on a Tecnai T12 (Thermo Fisher Scientific (formerly FEI), Hillsboro, Oregon, USA) transmission electron microscope at 100 kV acceleration voltage. The images were acquired with a F416 camera (Tietz Video and Image Processing Systems GmbH, Gilching, Germany) at 4096 × 4096 pixels using SerialEM software. Before taking an image, the sample was automatically exposed to the beam for 0.5–2 s to reduce drift. The exposure time (between 0.5 and 1.5 s) was subdivided into 3 subframes which were then automatically aligned by cross correlation and added up in order to form the final image in order to reduce image blurring by sample drift.

**Cryo TEM**
JCVI-Syn3A cells were adapted to FBS, POPC + cholesterol, and D.PC + cholesterol diets and harvested at between $OD_{600}$ 0.05–0.2 (mid-exponential growth phase) at volumes to achieve 1ODU of cells for each diet. Shortly before freezing, cells were spun down and resuspended in 1 mL Mycoplasma wash buffer without glucose (20 mM HEPES, 200 mM NaCl). Quantifoil 2.1 copper 200 mesh grids were cleaned with chloroform before usage. Glow discharging was performed on a PELCO Easy glow for 30 s, 15 mA on both sides of the grid. 2 μl of sample being added to both sides of the grid, 1 μl of gold (Protein A gold, PAG 10, pre-diluted 1:25 was added to the carbon side only. Cryo fixation was performed by plunge freezing in liquid ethane using a Leica GP with humidity chamber set to 21 °C, a humidity of 98% and a blotting time of 5 s from the back of the grid with Whatmann paper No. 1. Frozen grids were stored in liquid nitrogen[80].

Cryo-electron tomography was done on a Titan Halo transmission electron microscope with field emission gun electron source and a Gatan K2 Summit direct electron detector at 300 kV with an energy filter using a slit width of 20 eV. Full grid overview was acquired with SerialEM by automatically acquiring and stitching low-magnification (×210) images. Tilt series were taken with SerialEM on areas of interest at ×30,000 nominal image magnification, calibrated pixel size of 2.36 Å (super-resolution mode) and 2° increments with a bidirectional tilt scheme from 20° to −58° and from 22° to 58°. The acquisition was done with a defocus target of 5 μm and the accumulative dose was 80 – 90 e− per Å2 per tomogram. Images were acquired in dose fractionation mode with frame times between 0.10 and 0.25 s. Correction of the sample motion induced by the electron beam was done with Motion-Cor. Tomogram reconstruction was performed using Etomo from IMOD 4.11.18 using weighted back projection. Contrast transfer function curves were estimated with Ctfplotter and corrected by phase-flipping with the software Ctfphaseflip, both implemented in IMOD. Dose-weighted filtering was performed by using the mtffilter implemented in IMOD. To enhance the contrast of macromolecular structures and fill up missing wedge information IsoNet was used. Visualization of tomograms and averaged electron density maps was performed in 3dmod from IMOD[81]. Computer visualization of three-dimensional image data using IMOD. J. Struct. Biol. 116:71–76.); rendering of isosurfaces and structure fitting was performed using UCSF ChimeraX 1.8, developed by the Resource for Biocomputing,

Visualization, and Informatics at the University of California, San Francisco, with support from National Institutes of Health R01-GM129325 and the Office of Cyber Infrastructure and Computational Biology, National Institute of Allergy and Infectious Diseases.

## Statistical Analysis

For growth rate calculations in Figs. 3, 4, and 5 the number of biological and analytical replicates is given in Supplementary Data S1. For the propidium iodide assays in Fig. 5 there are three biological replicates each with three analytical replicates. For the FDA assay in Fig. 5 there are three biological replicates each with three analytical replicates. For the c-laurdan assay in Fig. 6 there are three biological replicates each with three analytical replicates. Statistical significance was calculated with an unpaired t-test. All error bars indicate the mean ± the standard deviation.

## Materials

SP4 (For 1 L):

PPLO (3.5 g) (Becton, Dickinson, and Company product no. 255420)

Tryptone (10 g) (Sigma product no 70169)

Peptone (5 g) (Sigma product no 70176)

20% Glucose (25 mL) (Ross product no. X997.2)

20% Yeastolate (10 mL) (Becton, Dickinson, and Company product no. 255772)

15% Yeast Extract (35 mL) (Roth product no. 2904.3)

70 g/L BSA (85 mL) (Sigma product no. A7030) or FBS (170 mL) (Biowest Product no. 5181H-500)

400,000 U/mL Penicillin G-sodium salt (2.5 mL) (Roth product no. HP48.1)

10 mg/mL L-Glutamate (5 mL) (Roth product no. HN08.2)

Sodium Bicarbonate (1.04 g) (Honeywell product no. 71630)

CMRL (4.9 g) (US Biological Lifesciences product no. C5900)

Phenol Red (11 mg) (Sigma product no. P3532)

Lipids:

Cholesterol (Avanti product no. 700100)

Diether POPC (Diether 1-palmitoyl-2-oleoyl-glycero-3-phosphocholine) (Avanti product no. 999983)

Diether DPPC (Diether 1-palmitoyl-2-palmitoyl-glycero-3-phosphocholine) (Avanti product no. 999992)

Diether DOPC (Diether 1-oleoyl-2-oleoyl-glycero-3-phosphocholine) (Avanti product no. 999991)

Diether POPG (Diether 1-palmitoyl-2-oleoyl-glycero-3-phosphotdylglycerol) (Avanti product no. 999973)

POPC (1-palmitoyl-2-oleoyl-glycero-3-phosphocholine) (Avanti product no. 850457)

DPPC (1-palmitoyl-2-palmitoyl-glycero-3-phosphocholine) (Avanti product no. 850355)

DOPC (1-oleoyl-2-oleoyl-glycero-3-phosphocholine) (Avanti product no. 850375)

POPG (1-palmitoyl-2-oleoyl-glycero-3-phosphotdylglycerol) (Avanti product no. 840457)

DPPG (1-palmitoyl-2-palmitoyl-glycero-3-phosphoglycerol) (Avanti product no. 840455)

DOPG (1-oleoyl-2-oleoyl-glycero-3-phosphoglycerol) (Avanti product no. 850475)

POPE (1-palmitoyl-2-oleoyl-glycero-3-phosphoethanolamine) (Avanti product no. 850757)

DPPE (1-palmitoyl-2-palmitoyl-glycero-3-phosphoethanolamine) (Avanti product no. 850705)

DOPE (1-oleoyl-2-oleoyl-glycero-3-phosphoethanolamine) (Avanti product no. 850725)

Oleic Acid (cis−9-Octadecenoic acid) (Sigma product no. 01383)

Palmitic Acid (1-Pentadecanecarboxylic acid) (Sigma P0500)

Enantiomeric POPC (3-palmitoyl-2-oleoyl-glycero-1-phosphocholine) (Avanti product no. 850855)

16-0 Cardiolipin (1′,3′-bis[1,2-dipalmitoyl-sn-glycero-3-phospho]-glycerol) (Avanti product no. 710333)

Assays:

FDA (Sigma product no. F7378)

Fluorescein (Fluka product no. 28803)

Propidium Iodide (Sigma product no. 537059)

C-Laurdan (Stratech Scientific Ltd. product no. T0001-SFC-1)

Cells:

JCVI-Syn3A-*mCherry* (From Telesis Bio)

*Mycoplasma mycoides subspecies capri strain GM12* (From Telesis Bio)

Machines:

TECAN Spark 20 M with Te-Cool cooling module

Biomek i7 Automated Workstation

DeNovix DS-11 FX +

PicoQuant FluoTime 300 High Performance Fluorescence Lifetime and Steady State Spectrometer

## Reporting summary

Further information on research design is available in the Nature Portfolio Reporting Summary linked to this article.

## Data availability

The lipidomic datasets generated in this study have been uploaded to the Zenodo repository and can be found at https://doi.org/10.5281/zenodo.13817894. Due to proprietary restrictions imposed by the data provider, Lipotype GmbH, raw lipidomics instrument files cannot be deposited in a standard metabolomics repository, as they contain commercially sensitive information. The TEM data has been deposited in the EBI Biolibraries repository and can be found at https://doi.org/10.6019/S-BSST1651. The cryo-EM data has been deposited in the Electron Microscopy Data Bank (EMDB) within the pDBE database, and can be found with the accession codes: EMD-51614, EMD-51606, EMD-51607, EMD-51608, EMD-51609, and EMD-51610. Source data are provided with this paper.

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

## Acknowledgements

The authors wish to thank the Saenz lab; Lisa Junghans, Ha Ngoc Anh Nguyen, and Tomasz Czerniak for discussions and feedback through the process, and for help with experimental designs; the JCVI, and specifically John Glass and Kim Wise, for providing *M. mycoides* and JCVI-Syn3A and for further feedback, including manuscript comments; Helena Jambor for assisting with figure design; and Jennifer Katz-Brandoli for assisting with robot runs and figure construction. We would like to thank Telesis Bio, Inc. for allowing us to use JCVI-syn3A and Lipotype GmbH for lipidomic analyses. We would like to thank the Electron Microscopy Facility of the MPI-CBG for their support. We additionally wish to thank Michaela Wilsch-Bräuninger from the MPI-CBG for insights regarding EM images, Mareike Jordan from the Von Appen Lab for assistance in reconstructing tomograms, as well as Adrian Nievergelt from the Gaia Pigino Group at the Human Technopole Milano for his help processing cryo-EM tomograms with IsoNet. This work was supported by the B CUBE of the TU Dresden, a German Federal Ministry of Education and Research BMBF grant (to J.S., project 03Z22EN12), and a VW Foundation "Life" grant (to J.S., project 93090).

## Author contributions

IJ and JPS conceptualized the experiments for this manuscript. IJ carried out the experiments, apart from the lipidomic data in Fig. 2B (2FA) which was acquired by NS. Electron microscopy experiments were conceptualized between IJ, JPS, PK, and AA and carried out jointly by IJ and PK, with assistance from the Electron Microscopy Facility of the MPI-CBG. IJ and JPS wrote the manuscript.

## Funding

## Competing interests

The Authors declare no competing interests.
