## [Transparent Peer Review file · Nature Communications]

A tuneable minimal cell membrane reveals that two lipid species suffice for life

Corresponding Author: Dr James Saenz

Version 0:

Reviewer comments:

Reviewer #1

(Remarks to the Author)

In their manuscript, Justice & Saenz present a pioneering approach to minimise the lipid composition of a living cell by utilising the ability of *Mycoplasma mycoides* to rely solely on lipids and fatty acids provided by the growth medium. The difficulty of controllably modifying and simplifying the lipid composition in vivo has been a major challenge for the membrane biology field, resulting in a large experimental gap between the models used in vivo, in vitro and in silico studies. As a result, the translation of in vitro/in silico findings to the context of a living cell has been challenging and frequently impossible. The approach presented in this manuscript is a major step forward in closing this gap and *Mycoplasma mycoides* now appears to provide a superb lipid-tuneable in vivo model for studying physical and chemical properties of the membrane directly in the context of a living cell. The presented experimental evidence that heterochiral lipidomes can support vital cellular functions is a nice demonstration of the type of well-controllable experiments the new model enables. I see potential of this model to become a key research tool bridging in vivo and in vitro membrane biology and contributing to better fundamental understanding of biological membranes.

Minor Comments:

Title: "two lipid species" rather than "two lipids" is perhaps better.

Abstract: I think the authors rather undersell their study here a bit. Yes, there can be new directions in bioengineering, but the approach is certainly even more clearly valuable for fundamental research on function of biological membranes.

Lines 38-39, 318-320: While far from the degree of tight control enabled by the method/model here, the papers listed below describe previous approaches to modify bacterial fatty acid composition to its study biological consequences. It would seem appropriate to mention them here, in addition to current references 5 and 6 (I am not sure why the reference 7 is included, which discusses cell wall stress regulatory networks)

Cronan Jr JE, Gelmann EP (1973) An estimate of the minimum amount of unsaturated fatty acid required for growth of *Escherichia coli*. *J Biol Chem* 248: 1188–1195

Boudreaux et al (1981) Biochemical and genetic characterization of an auxotroph of *Bacillus subtilis* altered in the acyl-CoA:acyl-carrier-protein transacylase. *Eur J Biochem* 115: 175–181

Gohrbandt et al (2022) Low membrane fluidity triggers lipid phase separation and protein segregation in living bacteria. *EMBO J*. 41(5):e109800.

Line 147-148: "Critical for growth" is a bit overstated given the two-fold reduction in growth speed observed in the absence. Please rephrase.

Line 214-215: Would the presence of an undiscovered lipase in JCVI-Syn3A-genome not suggest that such activity is in fact essential, which is surprising given the data presented in this manuscript. Perhaps the authors can extend upon this apparent discrepancy a bit?

Lines 216-220: It's not sufficiently clear what the authors mean with "not all cells for a given lipid diet survived the growth protocol". Do you mean in some cases the cultures grew and on other cases they did not? Please rephrase for clarity. Also, since viability was not tested, the authors should limit their statement to ability/inability to grow here (and not discuss viability).

Line 238-241 and 343-345: Formally (and admittedly theoretically), the lipid divide is also consistent with the possibility that LUCA between bacteria and archaea is more ancient than evolution of cellular membranes. The finding that heterochiral membranes support life eliminates the need to speculate about adoption of membranes composed of the different enantiomers as separate evolutionary events. The authors could consider discussing this implication as well.

Line 255-261: To interpret the hypoosmotic shock data, it would be critical to know whether the used *Mycoplasma mycoides* strain (and JCVI-Syn3A) encodes mechanosensitive channels that open in response to excess turgor (the activity of which is known to be intricately lipid-dependent) or does not encode, in which case a membrane rupture in response to excess turgor is occurring. At least some *Mycoplasma* species encode mechanosensitive channels according to Booth et al (2015) The evolution of bacterial mechanosensitive channels. *Cell Calcium*.

If mechanosensitive channels are not present, stating this would improve clarity and strengthen reasoning, allowing the interpretation that the observed change in lysis reflect changes in membrane physical stability. If mechanosensitive channels are present, however, a more nuanced discussion including the possibility of altered channel activity upon altered lipid composition would be needed.

Material and Methods:

-Several of the sections (PI, FDA and Laurdan) are written in a "protocol" formal with incomplete sentences. Please harmonise with the style used in other sections.

-Information about the biological vs technical replicate, and the used tests to indicate the statistical significance of the shown data is missing.

-Which TECAN fluorometer was used?

Legend S5: replace grtrowth with growth.

Reviewer #2

(Remarks to the Author)

In this study, the authors take advantage of the polar lipid scavenging capacity of *Mycoplasma* strains to ask fundamental questions about the limits of lipid composition in cells. They focus on two questions: 1) what is the minimal number of lipids that can support a cell and 2) what are the effects of changing the chirality of phospholipids. They take advantage of ether-linked phospholipids to avoid lipid degradation by native lipases and subsequent recycling that drives de novo lipid synthesis. They then characterize the effects of different feeding strategies on the resulting lipidome and growth of the cells.

The approaches and questions the authors are address are novel and would be of interest to a broad audience. The approach is creative and there are a couple of striking results. However, I think the study would be strengthened by a deeper analysis of cell physiology, even if the number of different directions are fewer. I also think some of the findings are not fully interpreted and the overall written and graphical presentation could be significantly improved. My specific comments are below.

1) The physiological analyses of the effects of lipid manipulation in this current manuscript are quite modest. While growth rates differences (here largely indirectly measured by medium acidification) are notable, better connection to changes (or lack thereof) in membrane morphology (e.g. by electron microscopy) or function of specific membrane proteins (ETC components?) would really enhance the study. As is, the results are intriguing, but could be a lot more informative if there were more techniques utilized cells growing on the different lipid states.

2) In their results and discussion, authors don't consider the important biophysical differences between lipid classes. Cholesterol is a very different lipid than PC, and cardiolipin also has unique properties. The authors should better describe these differences and how they might support these minimal membranes. Here they are discussed as purely chemical classes, without the breadth of functional knowledge that is known about them.

3) Relevant to the above, many papers by the Dowhan lab on manipulation of *E. coli* headgroups composition are not cited or discussed. These previous studies altered headgroup composition through genetics have strong relevance to the specific lipids looked at here and specific lipid interactions that are mentioned here (e.g. PC and CL).

4) The lipidomics method description looks to be a generic one provided by LipoType. I don't see any information about analysis of cholesterol nor the diether PC, which is a very unusual lipid that is not included in most routine quantifications. It would also be helpful if the authors provide the lipidomics dataset as an added data table.

5) The authors' claims that chain headgroup heterogeneity being more important than that for headgroup is based on comparing diether PG + cholesterol and 2 FAs + cholesterol, given that both accumulate the same major PL classes (PG + CL). However, this claim glosses over many differences in the lipidomes here: one has natural ester linkages, major differences in the headgroup stoichiometries, the presence of substantial DAG. The actual acyl chain diversity in the FA-fed comparison is also not provided.

6) Wouldn't looking at chain variation be best addressed by feeding multiple diether species with defined compositions? Perhaps these are not available synthetically, but it would address the overall question more than comparing the fatty acid fed cells with the diether PC/PG cells.

7) In the growth rate plots, no information is provided on what the replicates are and what statistics are being used in the plots. In general, the presentation of the figures could overall be improved for aesthetics, clarity, and consistency.

8) The stated finding that racemic mixtures of POPC perform more poorly than the entPOPC would be quite amazing and, since it could imply that racemic membranes would be selected against through an undefined mechanism. Looking at the SI growth data by OD, however, I am not sure there is a robust difference between the two. This made me somewhat concerned about the phenol red measurement of growth and how accurate it's representing growth differences.

9) It's not really clear in the manuscript what insights the JCVI-Syn3A experiments are providing. It seems to just act as a poorer growing background and the results for it as mycoides are similar when they are shown.

Reviewer #3

(Remarks to the Author)

In this manuscript, I. Justice and J.P. Saenz use two model organisms, *Mycoplasma mycoides* subsp. *capri* and its derivative, the minimal cell JCVI-syn3A, to explore the effect of various lipid diets on membrane composition and cell fitness. The authors take advantage of the fact that mycoplasmas have a very limited lipid metabolism complexity and contain only one cellular membrane, in opposition with other model bacteria such as *E. coli* or *B. subtilis*. They specifically investigate the impact of head group, acyl chain diversity, ester vs ether linked phospholipids, and lipid chirality using various methods such as thin layer chromatography, lipidomics, growth assays and membrane permeability assays. By doing so, the authors report the successful creation of a minimal cell with only two lipids in its lipidome.

First of all, I must say that the creation of a minimal organism with only two structure is indeed very exciting, and that the topics explored in this manuscript should be of great interest for most biologist in the field. Using ether phospholipids was a very elegant and ingenious way found by the authors to bypass lipid scavenging in mycoplasmas, and testing the compatibility of G1P enantiomer in bacteria is particularly interesting. However, the study reported by I. Justice and J.P. Saenz has some important flaws that need to be addressed before considering this manuscript for publication. I don't think the results currently included in the paper adequately support the conclusions made by the authors. The growth rate results are particularly problematic; from the results, it is difficult to conclude if the cells are really growing or simply surviving for a certain lapse of time. Raw growth curves should be provided for each condition tested as well as additional controls (e.g. cells in SP4 + FBS) to help readers interpret the results. From what I understand, the cells grown in defined diets display growth rates of 0.1 at best, which corresponds to a doubling time around 8 hours (vs ~60 min for *M. mycoides*). Since the growth of mycoides is severely impaired, I think that the conclusion of this paper should be revisited. I also found myself very disappointed by the serious lack of finish of this study. Supplemental tables S1 and S2 are missing, supplemental figure legends have many errors, parts of figures are missing, replicate numbers and statistical tests are lacking, many abbreviations, details, and labels are lacking or not uniform, and there are many errors in the text. Altogether, these issues make the reading process difficult and somewhat frustrating. The authors should revise their entire manuscript before proceeding to a new submission.

Considering this, I regret to say that I cannot consider this study for publication in its current form. If the authors submit a revised and improved version of this study, I will be happy to review their work again. I hope that the specific points raised below will help the authors to improve their manuscript.

Major points:

1. Figure 1: cardiolipins are not displaying correctly on the figure if we compare to the legend (one notched diamond vs two linked diamonds). The different CLs at point 5 are also lacking on the figure (no molecule is illustrated). Please fix. Also, at line 96 we can read "or reinserted into the membrane (Fig. 1; 6)", but in the figure point 6 refers only to the FFA pool. I think we should read "(Fig. 1; 6, 7)".
2. Figure 2a: since the TLC aspect is modified according to Figure S4, it is very difficult to discriminate PC from SM and lyso PC, and vice versa. For example, in the text it is stated cells unadapted to POPC + cholesterol (p=1) show only traces of SM, but looking at the figure this seems to be the opposite. I think the full TLC length should be displayed to facilitate interpretation, along with a standard as in Figure S4. In addition, I think the right part of the figure (Diether PC + Cholesterol diet) is currently not mentioned anywhere in the text.
3. Figure S4, related to Figure 2 and 3: I think it is a great idea to give full TLCs results in supplementary materials. However, this figure looks like a lab notebook; labels and terms are not uniform (e.g. std vs standard, Lyso PC vs Lysophospholipid, etc.), there are some cropped labels (...terol" on TLC #10) and some ambiguities ("cells" refers to *mycoplasma mycoides* or syn3??), and the figure legend is full of errors (Lipids with a cap l in title, Cardiolipid instead of cardiolipin at line 1, D. POPC

repeated twice at last line, lacking abbreviation description such as DOPE, DPPG, etc.). On top of that, essential information is lacking to allow correspondence with TLCs displayed in Figure 2 and 3, or results given in Figure S1, which is critical in the circumstances. A label above each TLC could help figure out which TLC goes with which experiment described in main text. Overall, this makes the interpretation of TLCs placed in main really painful. Please revise, correct, and make labels more uniform.

4. Legends of supplementary figures are filled with errors and are lacking abbreviations to clearly understand the figures. Here is a non-exhaustive list of errors that I found by quickly reading them:

- Figure S1: "specificallyby" lacking a space at line 2; "Deteermined" at line 3; "occurring" instead of "occurring" at line 5
 - Figure S3: "occurring" instead of "occurring" at last line
 - Figure S4: see my previous comment
5. Lines 123 and 408: where are the supp Table S1 and S2? These files seem to be missing from the current manuscript.
6. Line 128: what are the 28 lipid structures identified by MS? Could you give the list in a supp table?
7. Line 148: where are those growth rate results? If they correspond to D. PC in Figure 3f, please specify.
8. From the growth results shown in Figure 3f and Figure S5, it is really difficult to confirm that the cells are really growing in the different media and not just "surviving" for a certain period. For example, in Figure 3f, the results represent a growth rate based on phenol red color change. If we convert this growth rate to a doubling time, this gives a doubling time of ~8 hours ($GR = \ln 2/DT$), which is greatly lower than the doubling time of *M. mycoides* in SP4 medium (~60 min). If I misinterpreted the y-axis of this figure and it instead represents a medium pH change, then where is the graph showing the correlation between medium pH and OD560nm for their media? Anyhow, the authors should provide raw growth curves of all growth conditions compared in Figure 3f and add a SP4 + FBS control to clarify this.
9. Figure S5, S6, and Figure 5: I am wondering how reliable OD600nm data are for monitoring the growth of *M. mycoides*. Most Mollicutes are too small to be accurately measured at 600nm, and I do not recall any paper using that method to follow the growth of *M. mycoides*. Most studies used phenol red (as the authors also did) or DNA quantification using fluorescence staining as described in Breuer et al. 2019 and Hutchison et al. 2016, for example. In fact, considering previous studies (e.g., Meur et al. 1989), turbidity only appears after prolonged incubation periods (>18h) and is not correlation with viable cell counts but rather indicates the accumulation of cell agglomerates (i.e. dead cells). Can the authors comment on this? I think that a correlation between CFUs and OD600nm data should be made to ensure that measuring OD600nm is a reliable method to follow to growth of *M. mycoides*.
10. Related to my previous point: in Figure S5b, I'm not sure what these units mean (e.g., 0.250, 0.251, 0.252). Am I missing something here? These log axes and units make this graph uninterpretable. If OD600nm is indeed a reliable method to follow the growth of *M. mycoides*, we should see a clear curve using non-transformed data and linear axes. This gives the impression that OD600nm and A562 are correlated only for a very small range of values, and not necessarily during the exponential growth phase. In Figure S5A, where are these points located on the growth curve? Once again, the transformation of data makes the interpretation difficult. OD600 and A562 could be correlated for a very small range of values (which could be the case here considering the very small A562 variations plotted on the graph) during the stationary or death phase and gives the impression that both methods are reliable to measure growth.
11. Line 153: did your lipidomics data confirm that cells only contain 16:0/18:1 diether PC configurations? Could you provide a table of analyzed lipidomics data (same for the 28 lipid structures at line 128)?
12. Figure 4a: I do not see the interest of showing how the JCVI-syn3A genome was segmented and built here. Is there a relation between a certain genome segment and specific results reported in the text? Lines 199 to 204 could be removed or moved in the introduction section.
13. Lines 216-217: "To not bias the growth rates, these cells were removed from the growth rate analysis". How do you remove dead cells from cultures? Do you mean replicates?
14. Figure 4b and 4c: once again, given the very low growth rate reported, I would be very interested in seeing the raw growth curves of these conditions, especially since the authors reported that some replicates were also not growing in many conditions. How do these results compare to a JCVI-syn3A grown in SP4 + FBS? Without this comparison, I think the authors should mitigate their statement at lines 226-228.
15. How many replicates did the authors perform for growth experiments shown in Fig. 3f, 4c, and figure 5? I could not find this information anywhere in the text.
16. Figure 5b and 5c: how were these growth rate calculated? Phenol red or OD600?
17. Histograms in figures 6 and 5b, c: how many replicates? Do bars and error bars indicate mean + SD, median + SEM, or something else? Please specify.
18. The information about statistical tests seems to be lacking in the entire manuscript. For example, what test did the authors use in Figure 5b, c, and 6?
19. Supplementary Figures S2 and S3 are not cited anywhere in the text.

Minor points:

1. Line 40: *Mycoplasma*"s" are a class? Or if you are referring to the genus, italicize and put singular.
2. Line 43, 369, and 382: italicize "in situ", "in vitro", and "in vivo".
3. Line 50: add a dot after "subsp" ("subsp.").
4. Line 51: do not italicize the strain name GM12.
5. Line 54-55: kind of difficult to read. Please reformulate to clarify.
6. End of line 91: missing dot at the end of sentence.
7. Line 109: CL was not defined earlier in the text. Define at line 95 or 97 before using the abbreviation.
8. Line 111: Define POPC before using the abbreviation.
9. Line 116: Define SM before using the abbreviation. Many lipid abbreviations are not defined before using in the text. Revise the entire manuscript.
10. Figure 2: to facilitate interpretation, please define lipid abbreviations in all Figure legends, including supplemental figures.

11. Line 184: replace “do no” by “do not”.
12. Line 219: there is a comma missing in this sentence.
13. Line 335-336: italicize “*Mesoplasma*”. I would also cite the original description of *Mesoplasma florum* L1 to support that point (Mccoy et al. 1984).
14. Line 357 and 359: italicize “*E. coli*”.
15. Line 392: “t” in “the Saenz” lab is underlined.
16. Lines 426-427: E0 or Eo? Different between equation and text.
17. Figure 4c legend: since D. PC comparison is not significant, change “same diets” for “D. PC diet”.
18. Lines 523-549: the font size and type seem to be varying in this paragraph. Same for the liposome preparation paragraph. Please make uniform.

Version 1:

Reviewer comments:

Reviewer #1

(Remarks to the Author)

The authors have done a solid job in responding to my comments (and those of the other reviewer), resulting in a significantly improved manuscript. I have no further concerns and I am happy to endorse the publication of the manuscript.

Reviewer #2

(Remarks to the Author)

The revisions by Justice et al substantially improve this interesting manuscript. I now support its publication, with the following comments:

- 1) I especially appreciate the addition of electron microscopy data and tomography on JCVI-Syn3A. I only wish this type of analysis was done more broadly, e.g. on *M. mycoides* minimal lipid systems, to understand if the intracellular vesicles are a result of the minimal host or the lipids.
- 2) As the methods make clear now, the diether-PC was quantified using a PC standard. It does not appear that the response factor for PC was calibrated with that for diether-PC. Thus, analysis of diether-PC is not quantitative, but more of an estimate. This does not impact any of the paper's conclusion, but should be noted in the methods and/or caption for Figure 2.

Reviewer #3

(Remarks to the Author)

I thank the authors for revising and correcting their manuscript. I have no further comments.

REVIEWER COMMENTS

Reviewer #1 (Remarks to the Author):

In their manuscript, Justice & Saenz present a pioneering approach to minimise the lipid composition of a living cell by utilising the ability of *Mycoplasma mycoides* to rely solely on lipids and fatty acids provided by the growth medium. The difficulty of controllably modifying and simplifying the lipid composition in vivo has been a major challenge for the membrane biology field, resulting in a large experimental gap between the models used in vivo, in vitro and in silico studies. As a results, the translation of in vitro/in silico findings to the context of a living cell has been challenging and frequently impossible. The approach presented in this manuscript is a major step forward in closing this gap and *Mycoplasma mycoides* now appears to provide a superb lipid-tuneable in vivo model for studying physical and chemical properties of the membrane directly in the context of a living cell. The presented experimental evidence that heterochiral lipidomes can support vital cellular functions is a nice demonstration of the type of well-controllable experiments the new model enables. I see potential of this model to become a key research tool bridging in vivo and in vitro membrane biology and contributing to better fundamental understanding of biological membranes.

We want to thank the reviewer both for their detailed feedback and for their enthusiasm for our results. We have provided detailed responses to your comments below:

Minor Comments:

1. Title: "two lipid species" rather than "two lipids" is perhaps better.

Excellent suggestion, we have modified the title.

2. Abstract: I think the authors rather undersell their study here a bit. Yes, there can be new directions in bioengineering, but the approach is certainly even more clearly valuable for fundamental research on function of biological membranes.

We appreciate the reviewer's remarks and agree that we can go further in highlighting the applications of this work fundamental research. To that end we have added the following sentences (highlighted below) in lines 21-23 of the abstract:

All cells are encapsulated by a lipid membrane which facilitates the interaction between life and its environment. How life exploits the diverse mixtures of lipids that dictate membrane property and function has been experimentally challenging to address. We introduce an approach to tune and minimize lipidomes in *Mycoplasma mycoides* and the Minimal Cell (JCVI-Syn3A) revealing that a 2-component lipidome can support life. Systematically reintroducing phospholipid features demonstrated that acyl chain diversity is more critical for growth than head group diversity. By tuning lipid chirality, we explored the lipid divide between Archaea and the rest of life, showing that ancestral lipidomes could have been heterochiral. Our approach offers a tunable minimal membrane system to explore the fundamental lipidomic requirements for life, thereby extending the concept of minimal life from the genome to the lipidome.

3. Lines 38-39, 318-320: While far from the degree of tight control enabled by the method/model here, the papers listed below describe previous approaches to modify bacterial fatty acid composition to its

study biological consequences. It would seem appropriate to mention them here, in addition to current references 5 and 6 (I am not sure why the reference 7 is included, which discusses cell wall stress regulatory networks)

Cronan Jr JE, Gelmann EP (1973) An estimate of the minimum amount of unsaturated fatty acid required for growth of *Escherichia coli*. *J Biol Chem* 248: 1188–1195

Boudreaux et al (1981) Biochemical and genetic characterization of an auxotroph of *Bacillus subtilis* altered in the acyl-CoA:acyl-carrier-protein transacylase. *Eur J Biochem* 115: 175–181

Gohrbandt et al (2022) Low membrane fluidity triggers lipid phase separation and protein segregation in living bacteria. *EMBO J.* 41(5):e109800.

We thank the reviewer for highlighting these relevant papers, which are indeed fundamental to the development of an approach for modifying and controlling membrane composition through the addition of exogenous lipids. We have included the suggested citations and expanded the introduction to better address the significance of bacteria as models for understanding lipidome complexity. Additionally, we agree with the reviewer's assessment that the original citation 7 was not relevant to this discussion, we had intended a different citation from the Helmann group about reducing the lipid synthesis capacity of *B. subtilis*. We have now replaced it with the correct citation. The specific edits are detailed below (lines 32-45):

Introduction

Cell membranes are complex and responsive systems that serve to protect and mediate interactions of life with its environment. A large part of this molecular complexity is due to the diverse panel of lipids that make up the lipidome and which ultimately determine the form and function of the membrane. The complexity of cellular lipidomes can be staggering, from tens of unique structures in bacteria¹, to hundreds in eukaryotic organisms². How life has evolved to utilize such complex mixtures of lipids to build cellular membranes remains an active area of exploration^{1,3-6}. While synthetic membranes can be constructed using a single lipid species, the minimal number of lipid species required for a functional cell membrane remains undetermined. Identifying a minimal viable lipidome would provide a critical starting point for elucidating the combinations of lipid structures that are essential for membrane integrity and function, offering insights into the fundamental chemical and physical requirements of cellular life.

One approach to studying lipidome complexity is to experimentally manipulate lipidome composition and observe effects on cell fitness (e.g. growth). Bacterial model organisms have proven to be excellent systems for tuning lipidome composition by genetically disrupting lipid biosynthesis⁷. For example, early work with *Escherichia coli* mutants deficient in fatty acid synthesis explored the role of acyl chain unsaturation for cell growth⁸, and a similar approach with *Bacillus subtilis* mutants explored the role of branched acyl chains⁹. More recent work, employing approaches to tune lipid unsaturation, revealed the importance of homeoviscous adaptation for electron transport¹⁰ and demonstrated how low membrane fluidity can induce phase separation, impaired membrane potential, cell growth and division¹¹. However, for interpreting how lipidome composition and flexibility are affected by perturbed lipid synthesis, *E. coli*, like many bacteria, is complicated by the fact that it has multiple membranes (e.g. inner and outer). Changes in a single membrane are obfuscated by whole cell lipid extracts, and purifying specific membrane types is laborious, and can lead to substantial experimental error due to varying purity. Gram positive organisms, such as *B. subtilis*, that have only a single membrane are better models in this regard^{12,13}. However, traditional genetic approaches to tuning lipidome composition have so far not

afforded complete control over both lipid class composition and phospholipid acyl chain composition. Thus, a cellular model system in which lipidome complexity can be reduced in a systematic fashion has not yet been established.

4. Line 147-148: “Critical for growth” is a bit overstated given the two-fold reduction in growth speed observed in the absence. Please rephrase.

We agree with the reviewer that the phrase “Critical for growth” is confusing, and potentially misleading. We have changed the sentence to be more precise, reading (lines 184-185):

Minimizing the lipidome of *M. mycoides* to two lipids **resulted in a two-fold decrease in growth rates (Figure. 3e, diether PC condition compared to POPC condition).**

5. Line 214-215: Would the presence of an undiscovered lipase in JCVI-Syn3A-genome not suggest that such activity is in fact essential, which is surprising given the data presented in this manuscript. Perhaps the authors can extend upon this apparent discrepancy a bit?

We thank the reviewer for their insightful question and agree that the current discussion in the manuscript may appear to contradict our initial statement. In their work, "Bianchi et al. (2022) Toward the Complete Functional Characterization of a Minimal Bacterial Proteome," the authors used computational analyses to identify the structure and function of every gene in the minimal cell. One key finding was the extent to which proteins in JCVI-Syn3A exhibit moonlighting activity—a logical outcome in an organism operating with a reduced set of proteins. Consequently, the residual lipase activity observed in JCVI-Syn3A could plausibly result from the moonlighting function of another esterase. Regarding the essentiality of a lipase, the reviewer correctly notes that our results with diether lipids demonstrate that PLA1/2 lipase activity is not essential for cell survival. However, the observed reductions in growth rates on diets lacking lipase activity, compared to the POPC diet, underscore the critical role of acyl chain scavenging for organismal fitness. To clarify this point, we have made the following modification to the text (lines 277-281):

The presence of such lipase activity in a genomically minimal cell could mean that it is essential. Alternatively, it could also indicate that, while not essential itself, it is a secondary “moonlighting” activity of an enzyme with an essential activity⁴⁷. This implies that with the appropriate complement of lipids, the genome could be further minimized by eliminating genes involved in lipid remodeling activities.

6. Lines 216-220: It’s not sufficiently clear what the authors mean with “not all cells for a given lipid diet survived the growth protocol”. Do you mean in some cases the cultures grew and on other cases they did no? Please rephrase for clarity. Also, since viability was not tested, the authors should limit their statement to ability/inability to grow here (and not discuss viability).

We apologize for the confusion here and agree that the current wording is confusing and “viability” should not be claimed. After revising the manuscript, we realized that this data, while important supporting information, was not central to the main line of reasoning in the manuscript. Consequently, we moved this data to Supplementary Table S1, along with complementary data for *M. mycoides*, and we have included the following text in the table caption to clarify what we meant (which you correctly summarized as “not all cells for a given lipid diet survived the growth protocol”):

As a more minimal system, JCVI-Syn3A is a more fragile organism, and not all replicates for a given lipid diet survived the growth protocol. To not bias the growth rates, these replicates were removed from the growth rate analysis. An interesting feature was noticed, however, that a much higher fraction of replicates on diets that could not undergo acyl chain scavenging did not grow compared to replicates on diets that could, further illustrating the decreased fitness of cells unable to remodel acyl chain composition.

7. Line 238-241 and 343-345: Formally (and admittedly theoretically), the lipid divide is also consistent with the possibility that LUCA between bacteria and archaea is more ancient than evolution of cellular membranes. The finding that heterochiral membranes support life eliminates the need to speculate about adoption of membranes composed of the different enantiomers as separate evolutionary events. The authors could consider discussing this implication as well.

Thank you for this comment. This is an excellent point, and we have added the following text to expand on the implications of viable heterochiral membranes (lines 496-503):

Interestingly, this suggests that LUCA could have existed before the evolution of homochiral cellular membranes. This possibility would eliminate the need to view the lipid divide between bacteria and archaea as resulting from independent evolutionary events. Instead, it suggests the ancestral lipidome could have been heterochiral, consistent with a simpler path for the evolution of modern membranes. Additionally, the reduced fitness and increased membrane permeability resulting from heterochiral lipidomes suggests selective pressure against such membranes in ancestral organisms, favoring the evolution of the homochiral membranes characteristic of modern life.

8. Line 255-261: To interpret the hypoosmotic shock data, it would be critical to know whether the used *Mycoplasma mycoides* strain (and JCVI-Syn3A) encodes mechanosensitive channels that open in response to excess turgor (the activity of which is known to be intricately lipid-dependent) or does not encode, in which case a membrane rupture in response to excess turgor is occurring. At least some *Mycoplasma* species encode mechanosensitive channels according to Booth et al (2015) The evolution of bacterial mechanosensitive channels. Cell Calcium.

If mechanosensitive channels are not present, stating this would improve clarity and strengthen reasoning, allowing the interpretation that the observed change in lysis reflect changes in membrane physical stability. If mechanosensitive channels are present, however, a more nuanced discussion including the possibility of altered channel activity upon altered lipid composition would be needed.

We agree with the reviewer that knowledge of the presence (or absence) of mechanosensitive channels is useful for interpreting the osmotic shock data. As the reviewer aptly points out, many *Mycoplasma* have been found to have mechanosensitive ion channels. However, a search for mechanosensitive ion channels (both MscS and MscL) in the genomes of *M. Mycoides* and JCVI-Syn3A did not yield any annotated results. Neither did a BLAST of the sequences for a bacterial MscS and MscL against the proteome of either organism. Additionally, I referred back to the manuscript of Bianchi et al (2022) "Toward the Complete Functional Characterization of a Minimal Bacterial Proteome" to see if either their written predictions or alpha fold structures matched known mycoplasma Mscs. In all cases, no links pointing to either Msc was found. Furthermore, the manuscript "Brown et al (1991) Survival of Feline Mycoplasmas in Urine" details osmotic shock assays done with other mycoplasma species, with one of their findings being that two of the species they were looking at, *Mycoplasma gateae* and *Mycoplasma felis*, were more sensitive to osmotic shock than other mycoplasmas, and also lack Mscs. However, we cannot rule out the possibility

that undiscovered mechanosensitive channels exist in these organisms. If such channels are discovered, then our results would indicate that mechanosensitive channels are sensitive to lipid chirality. We have added further clarifying information to the manuscript, which can be found below (lines 362-368):

Furthermore, mechanosensitive ion channels which can protect cells from hypoosmotic shock have, to the best of our knowledge, not been reported in either organism, or annotated in the genomes, and are not present in all Mycoplasmas^{47,55}. Therefore, it is reasonable to cautiously interpret the susceptibility to lysis from hypoosmotic shock as indicative of membrane stability. However, we cannot rule out the possibility that there are undiscovered mechanosensitive channels, and this would suggest that mechanosensitive gating is sensitive to lipid chirality.

9. Material and Methods:

-Several of the sections (PI, FDA and Laurdan) are written in a “protocol” formal with incomplete sentences. Please harmonise with the style used in other sections.

We apologize for these formatting errors, we have corrected and rewritten the relevant parts of the materials and methods section.

-Information about the biological vs technical replicate, and the used tests to indicate the statistical significance of the shown data is missing.

We apologize for leaving that information out; It has now been added in abbreviated form to the appropriate figure captions, and in full form under the “Statistical Analysis” section of Materials and Methods.

-Which TECAN fluorometer was used?

A TECAN Spark was used. The text has been adjusted to include the full name.

Legend S5: replace grtrowth with growth.

We apologize for the error, it has been corrected.

Reviewer #2 (Remarks to the Author):

In this study, the authors take advantage of the polar lipid scavenging capacity of Mycoplasma strains to ask fundamental questions about the limits of lipid composition in cells. They focus on two questions: 1) what is the minimal number of lipids that can support a cell and 2) what are the effects of changing the chirality of phospholipids. They take advantage of ether-linked phospholipids to avoid lipid degradation by native lipases and subsequent recycling that drives de novo lipid synthesis. They then characterize the effects of different feeding strategies on the resulting lipidome and growth of the cells.

The approaches and questions the authors are address are novel and would be of interest to a broad audience. The approach is creative and there are a couple of striking results. However, I think the study would be strengthened by a deeper analysis of cell physiology, even if the number of different directions are fewer. I also think some of the findings are not fully interpreted and the overall written and graphical presentation could be significantly improved. My specific comments are below.

We are grateful for the reviewer's careful thoughts on our manuscript. We are excited that they believe the approach is novel and has the potential to be high impact, and we hope we can improve our clarity and presentation to better support our results.

1. The physiological analyses of the effects of lipid manipulation in this current manuscript are quite modest. While growth rates differences (here largely indirectly measured by medium acidification) are notable, better connection to changes (or lack thereof) in membrane morphology (e.g. by electron microscopy) or function of specific membrane proteins (ETC components?) would really enhance the study. As is, the results are intriguing, but could be a lot more informative if there were more techniques utilized cells growing on the different lipid states.

Our original intent was to focus on the remarkable result that two lipids are sufficient for life. Characterizing the physiological consequences of lipidomic minimization will be one of the major focuses in the lab for the coming years. We very much like your suggestion to include EM images to explore how lipidome minimization affects morphology in a genomically minimized cell, and we have included new results showing how lipidome reduction in JCVI-Syn3A leads to membrane invaginations and internal membrane vesicles in a subpopulation of cells. Our new transmission electron microscopy (TEM) and cryogenic electron microscopy (cryoEM) data reveal that cells grown on a simplified lipid diet predominantly maintain typical ovoid morphology, despite having a two-component lipidome. However, we observed notable subpopulations of cells displaying membrane invaginations and internal membrane-encapsulated vesicles (confirmed by cryoEM tomography), with the frequency of membrane invaginations increasing significantly with minimized lipidomes, suggesting impaired control of membrane curvature, and cell morphology, possibly due to the absence of CL and loss of optimal membrane curvature. This reinforces our broader conclusion that while a minimal set of lipids can support life, the complexity and diversity of the lipidome are likely critical for maintaining normal cell physiology and preventing structural aberrations. This work advances our understanding of the minimal requirements for life and highlights areas for future exploration, particularly in the context of how lipidome composition influences membrane curvature and cell division. We have added new data to Figure 4 (4c and 4d) Supplementary figure S6, and the following text has been added to the results section "Minimizing the lipidome of the Minimal Cell" (note: full resolution cryoEM tomograms and TEM images are being deposited on the EBI repository and will be made publicly available in the published version)(lines 292-326):

Previous studies reported that genome minimization in JCVI-Syn3.0 caused pleomorphic traits and abnormalities in cell division, which were rescued by reintroducing 19 genes, resulting in the creation of JCVI-Syn3A²¹. Given that a reduced lipidome could also impair cellular functions, we investigated whether lipidome minimization might lead to abnormal cell morphologies in JCVI-Syn3A. Transmission electron microscopy (TEM) images of JCVI-Syn3A grown on three different lipid diets—fetal bovine serum (FBS), POPC + cholesterol, and D.PC + cholesterol—revealed mostly typical ovoid cells (Fig. 4c, TEM overview images provided in Supp. Fig. 6a-g). However, two distinct morphological features were observed in subpopulations of cells: internal membrane-encapsulated vesicles and tube-like membranous structures connecting cells.

The tubules observed in fewer than 20% of cells, were less frequent in cells grown on POPC or D.PC compared to those grown on FBS. These structures are reminiscent of those seen in wall-less L-form bacteria, where they are hypothesized to function as an FtsZ-independent mechanism, possibly representing a primitive form of cell division⁴⁸. Further investigation is required to determine if the tubules observed in JCVI-Syn3A are functional or represent incomplete cell division. Notably, their higher prevalence in FBS-grown cells suggests that they are not directly related to lipidome minimization.

In contrast, the frequency of cells with membrane invaginations increased more than two-fold from ~15% in FBS-grown cells to nearly 40% in those grown on D.PC. In some instances, these invaginations appeared as distinct membrane-encapsulated vesicles within the cell, separate from the cell surface membrane. To confirm the presence of these internalized membrane vesicles, we used cryogenic electron microscopy (cryoEM) to construct whole-cell tomograms. Tomograms of JCVI-Syn3A from all three diets confirmed the presence of internal membrane-encapsulated vesicles (Fig. 4d, Supp. Fig. S6h-i). The lower electron density within these vesicles (pixel brightness), compared to the cytoplasm, indicates they result from membrane invagination, encapsulating extracellular fluid.

Interestingly, cells with reduced lipidomes were larger on average compared to those grown on FBS (~0.75 μm vs. 0.3 μm diameter respectively), as estimated semi-quantitatively from TEM images (Supplementary Fig. S6j). Cells with internal vesicles were particularly enlarged across all conditions (up to 1.5 μm average diameter). These results suggest that lipidome minimization leads to larger cell sizes and a higher frequency of membrane invaginations, indicative of impaired regulation of cell size and shape. The increased frequency of membrane invaginations could result from non-optimal membrane bending rigidity or intrinsic curvature from the loss of cardiolipin, and acyl chain diversity. Nonetheless, the fact that around half of the observed cells maintained normal morphology shows that even with just two lipid species, JCVI-Syn3A is capable of preserving typical cell morphology.

2. In their results and discussion, authors don't consider the important biophysical differences between lipid classes. Cholesterol is a very different lipid than PC, and cardiolipin also has unique properties. The authors should better describe these differences and how they might support these minimal membranes. Here they are discussed as purely chemical classes, without the breadth of functional knowledge that is known about them.

The reviewer here makes a very important point, and we are grateful for it. We aimed to keep the manuscript concise, and to this end our intent was originally in the first part of the manuscript to focus less on lipid structure-property relationships, and more on the path to minimizing the lipidome. But we agree that it does a disservice to the reader not to provide some context about the biophysical significance of the lipids. We have now added a paragraph in the results section introducing the biophysical differences between the lipid structural features discussed in the manuscript (shown below). We also added text throughout the results to more explicitly interpret biophysical implications where appropriate (lines 123-138):

The diverse lipid structures that *M. mycoides* can take up or synthesize ultimately determines the physical properties of their cell membranes^{37,38}. Phospholipid acyl chains can vary in terms of their length and degree of unsaturation, both influencing physical parameters such as membrane fluidity, thickness, and permeability. Phospholipid head groups such as PG and CL introduce a negative surface charge to the membrane, which can influence their interaction with peripheral membrane proteins. Conversely, PC is zwitterionic and introduces a neutral charge to the membrane surface. The geometric shape of phospholipids is also important, and determines whether a lipid spontaneously aggregates to form a bilayer or non-bilayer structure. For example, CL has four acyl chains and a relatively small headgroup giving it a conical profile. Cells have been shown to tune the abundance of such conical lipids to modulate the curvature and bending rigidity of their membranes³⁹. Sterols, which also do not form bilayers by themselves, play an important biophysical role in the membrane, including in modulating membrane fluidity, stability, facilitating liquid-liquid phase separation, and membrane asymmetry^{40,41}. By limiting the diversity of lipids that can be taken up or synthesized, we aimed to identify a minimal viable lipidome that can be used as an experimental platform in which lipid diversity can be systematically tuned in a living membrane.

3. Relevant to the above, many papers by the Dowhan lab on manipulation of *E. coli* headgroups composition are not cited or discussed. These previous studies altered headgroup composition through genetics have strong relevance to the specific lipids looked at here and specific lipid interactions that are mentioned here (e.g. PC and CL).

The oversight of the work of the Dowhan lab was short-sighted on our part, and we are grateful to the reviewer for their reminder of the work done in the Dowhan lab on the importance of cardiolipin. When writing this manuscript, we were focused more on the minimization of the lipidome, rather than a discussion of the specific lipid classes that were removed. However, we appreciate that many readers who work with lipids and membranes will be interested in some discussion about how our findings connect with pioneering work by Dowhan and colleagues. We have revised the introduction, results, and discussion to reference some of Dowhan's key works and engage in a broader discussion about the chemical differences between various lipid classes in the cell membrane, and the significance of the presence or absence of cardiolipin.

4. The lipidomics method description looks to be a generic one provided by LipoType. I don't see any information about analysis of cholesterol nor the diether PC, which is a very unusual lipid that is not included in most routine quantifications. It would also be helpful if the authors provide the lipidomics dataset as an added data table.

We thank the reviewer for this suggestion and apologize for the lack of clarity. The full lipidomics dataset has been added as Supplementary Table S3. We've edited the methods to specify how cholesterol and Diether PC were analyzed, and referenced Liebisch et al., 2006 for detailed background on the basis for cholesterol quantification. We confirmed with Lipotype that Diether PC was quantified using the PC standard (PC 17:0/17:0), the same procedure as for plasmalogens with ether-linked acyl chains (681-686):

Both MS and MSMS data were combined to monitor CE, DAG and TAG ions as ammonium adducts; PC, PC O-, and Diether PC as acetate adducts; and CL, PA, PE, PE O-, PG, PI and PS as deprotonated anions. MS only was used to monitor LPA, LPE, LPE O-, LPI and LPS as deprotonated anions; ceramide, hexosylceramide, sphingomyelin, LPC and LPC O- as acetate adducts and cholesterol as an ammonium adduct of an acetylated derivative⁷⁷.

5. The authors' claims that chain headgroup heterogeneity being more important than that for headgroup is based on comparing diether PG + cholesterol and 2 FAs + cholesterol, given that both accumulate the same major PL classes (PG + CL). However, this claim glosses over many differences in the lipidomes here: one has natural ester linkages, major differences in the headgroup stoichiometries, the presence of substantial DAG. The actual acyl chain diversity in the FA-fed comparison is also not provided.

The reviewer here makes a very important point. Our initial claim did not consider the chemical differences in head groups, ester vs. ether linkages, or the presence of the lipid intermediate DAG, leading to an oversimplification. To address this, we performed a shotgun lipidomic analysis of *M. mycoides* grown on the '2FA' diet (cholesterol, palmitate, oleate), now presented in Figure 2b. The lipidomic data revealed that DAG is indeed present but in very low abundance (1.8 mol%, Supplemental Table S3). The low abundance of DAG suggests that its impact on membrane physical properties is minimal. Additionally, because DAG, like the more abundant cardiolipin, is a conical-shaped lipid, DAG does not introduce a new structural feature to the lipidome, in that regard. The text has been revised accordingly (lines 218-239):

To understand the impact of phospholipid acyl chain diversity on cell growth, we grew cells on a diet of cholesterol and two fatty acids (palmitate and oleate), designated as '2FA'. On this diet, the phospholipidome is predominantly composed of PG and cardiolipin (Fig. 2b), allowing for the synthesis of phospholipids with various acyl chain configurations (e.g., 16:0/18:1 POPG, 16:0/16:0 DPPG, 18:1/18:1 DOPG for PG, and different permutations of PGs as substrates for cardiolipin synthesis). Although DAG and PA are present in low abundances (1.8 mol%, and 0.3 mol%, respectively, Supplemental Table S3), they are also both conical-shaped lipids, similar to cardiolipin. In that respect, their contribution to the membrane's physical properties is relatively small and does not introduce significant differences compared to what is already provided by the far more abundant cardiolipin. Therefore, while the 2FA diet exhibits comparable headgroup diversity to cells grown on D.PG, they exhibit greater acyl chain diversity. Growth rates on the 2FA diet were more than double those on D.PG and approached the growth rates of cells adapted to POPC (Fig. 3e). This suggests that for a lipidome with predominately two phospholipid headgroups, increased acyl chain diversity can rescue growth. An important consideration in comparing cells grown on the 2FA diet versus the D.PC + D.PG diet is that phospholipids with diether-linked acyl chains are not natural for this organism and could introduce a growth deficit. However, the similar growth rates observed in cells grown on POPC prior to adaptation, which could not yet synthesize PG and cardiolipin, compared to D.PC diets, suggest that reduced growth is not primarily due to the introduction of diether phospholipids. Acknowledging that differences in natural ester linkages and unnatural ether linkages, and the presence of DAG and PA—albeit in low abundance—may also influence the observed growth rates, these observations suggest that acyl chain complexity is an important factor in rescuing growth.

6. Wouldn't looking at chain variation be best addressed by feeding multiple diether species with defined compositions? Perhaps these are not available synthetically, but it would address the overall question more than comparing the fatty acid fed cells with the diether PC/PG cells.

This is in principle an elegant approach that we are discussing with a collaborator who can synthesize a variety of acyl chain configurations in ether lipids or other synthetic lipids that cannot be remodeled by mycoplasma or Syn3. However, it would require that these organisms have the capacity to selectively distinguish phospholipid species with different acyl chain configurations in a manner comparable to how they remodel acyl chains on phospholipids that are already in the membrane. We are actively working to characterize this, it is the topic of another student's project in the lab and would not be feasible to include

in this study in a reasonable time frame. But we are excited about the possibility to extend the approach reported in this manuscript to achieve even more control over lipidome composition in the future!

7. In the growth rate plots, no information is provided on what the replicates are and what statistics are being used in the plots. In general, the presentation of the figured could overall be improved for aesthetics, clarity, and consistency.

We apologize for leaving that information out; It has now been added in abbreviated form to the appropriate figure captions, and in full form under the “Statistical Analysis” section of Materials and Methods. Figures have also been edited for aesthetics, clarity, and consistency.

8. The stated finding that racemic mixtures of POPC perform more poorly than the entPOPC would be quite amazing and, since it could imply that racemic membranes would be selected against through an undefined mechanism. Looking at the SI growth data by OD, however, I am not sure there is a robust difference between the two. This made me somewhat concerned about the phenol red measurement of growth and how accurate it’s representing growth differences.

Phenol Red absorbance is generally considered to be a more standard approach for measuring mycoplasma growth than optical density (OD) of cells. We have found that both measures can be used and typically yield growth rate estimates that are proportional (Supplementary Figure S3b), however, they report different parameters. Phenol Red monitors pH change in the media, resulting from excreted products of metabolism, whereas OD monitors changes in cell density. Therefore, Phenol Red absorbance reflects metabolic activity, whereas OD reflects biomass. In this case, both measures support the conclusion that the introduction of entPOPC to the lipid diet leads to lower growth. However, while Phenol Red reports a large difference in growth between entPOPC and the racemic mixture, OD does not. In a separate study, we have measured the growth rates of these conditions using microcalorimetry, which is a direct measure of the heat flow generated by cells, and a quantitative measure of metabolic activity. The growth rates estimated from this approach corroborate the Phenol Red based growth estimates (shown below). We cannot include this data in this manuscript as it is part of a different study with a collaborator. However, this corroboration gives us confidence that the Phenol Red estimate is not an artefact. The discrepancy between OD and Phenol Red is an interesting observation that we intend to explore, but it goes beyond the scope of this study. It is important to consider also the other measures we have provided, such as membrane permeability to FDA, which is consistently much higher when cells are fed a racemic diet of POPC. Therefore, even without considering growth rate, there is an argument to be made that there could have been a selective pressure against heterochiral lipidomes. We have modified the text in this section to acknowledge the discrepancy between Phenol Red and OD growth rate estimates, and to clarify our interpretation (lines 351-356):

We also evaluated growth by measuring optical density, as a proxy for cell density (Supplementary Figure S7), which confirmed a reduction in growth with the introduction of entPOPC. However, in contrast, growth was similar for the entPOPC and racemic lipid diets. This difference likely reflects the fact that phenol red growth rate estimates reflect metabolic activity, which is not necessarily coupled with the production of cell biomass measured by optical density.

Growth data from microcalorimetry heat flow measurements:

9. It's not really clear in the manuscript what insights the JCVI-Syn3A experiments are providing. It seems to just act as a poorer growing background and the results for it as mycoides are similar when they are shown.

The reviewer raises a good point, and we appreciate the opportunity to clarify the insights provided by the JCVI-Syn3A experiments. JCVI-Syn3A, a genome-reduced form of *M. mycoides*, serves as a model system to examine the fundamental requirements of a membrane in a genomically minimal organism. By simplifying the Syn3A lipidome, our aim was to explore how few lipid components are required to maintain a functional membrane, thus advancing our understanding of the minimal requirements for cellular life. An additional practical advantage of Syn3A is its classification as a lower-risk organism (at least in Germany, where the study was performed). It lacks all genes known to be essential for pathogenicity in *M. mycoides*, allowing safer handling in laboratory environments. This safety aspect enabled us to perform CryoEM imaging on Syn3A, as our CryoEM collaborators lack facilities permitted for safety level 2 organisms like *M. mycoides*. We have added the following text to the discussion to motivate the significance of minimizing the lipidome in Syn3A (445-453):

By applying a targeted chemical approach to reduce the lipidome of JCVI-Syn3A, we demonstrated the feasibility of further simplifying the molecular composition of a genomically minimized organism. The observation that a minimal cell membrane can function with only cholesterol and one species of PC demonstrates that the fundamental requirements for life can be achieved with a remarkably simple lipid composition. For synthetic biology, this insight simplifies the challenge of designing synthetic cells, revealing the potential to create functional living membranes with minimal components. This work lays a foundation for future efforts to understand how minimal lipidomes can be optimized in synthetic and engineered biological systems

Reviewer #3 (Remarks to the Author):

*In this manuscript, I. Justice and J.P. Saenz use two model organisms, *Mycoplasma mycoides* subsp. *capri* and its derivative, the minimal cell JCVI-syn3A, to explore the effect of various lipid diets on membrane composition and cell fitness. The authors take advantage of the fact that mycoplasmas have a very limited lipid metabolism complexity and contain only one cellular membrane, in opposition with other model bacteria such as *E. coli* or *B. subtilis*. They specifically investigate the impact of head group, acyl chain diversity, ester vs ether linked phospholipids, and lipid chirality using various methods such as thin layer chromatography, lipidomics, growth assays and membrane permeability assays. By doing so, the authors report the successful creation of a minimal cell with only two lipids in its lipidome.

First of all, I must say that the creation of a minimal organism with only two structure is indeed very exciting, and that the topics explored in this manuscript should be of great interest for most biologist in the field. Using ether phospholipids was a very elegant and ingenious way found by the authors to bypass lipid scavenging in mycoplasmas, and testing the compatibility of G1P enantiomer in bacteria is particularly interesting. However, the study reported by I. Justice and J.P. Saenz has some important flaws that need to be addressed before considering this manuscript for publication. I don't think the results currently included in the paper adequately support the conclusions made by the authors. The growth rate results are particularly problematic; from the results, it is difficult to conclude if the cells are really growing or simply surviving for a certain lapse of time. Raw growth curves should be provided for each condition tested as well as additional controls (e.g. cells in SP4 + FBS) to help readers interpret the results. From what I understand, the cells grown in defined diets display growth rates of 0.1 at best, which corresponds to a doubling time around 8 hours (vs ~60 min for *M. mycoides*). Since the growth of *mycoides* is severely impaired, I think that the conclusion of this paper should be revisited. I also found myself very disappointed by the serious lack of finish of this study. Supplemental tables S1 and S2 are missing, supplemental figure legends have many errors, parts of figures are missing, replicate numbers and statistical tests are lacking, many abbreviations, details, and labels are lacking or not uniform, and there are many errors in the text. Altogether, these issues make the reading process difficult and somewhat frustrating. The authors should revise their entire manuscript before proceeding to a new submission.

Considering this, I regret to say that I cannot consider this study for publication in its current form. If the authors submit a revised and improved version of this study, I will be happy to review their work again. I hope that the specific points raised below will help the authors to improve their manuscript.

We greatly appreciate the reviewer's detailed and thoughtful feedback on our manuscript. We apologize for the copy editing errors present in the manuscript and thank you for this in-depth review.

Major points:

1. Figure 1: cardiolipins are not displaying correctly on the figure if we compare to the legend (one notched diamond vs two linked diamonds). The different CLs at point 5 are also lacking on the figure (no molecule is illustrated). Please fix.

Also, at line 96 we can read "or reinserted into the membrane (Fig. 1; 6)", but in the figure point 6 refers only to the FFA pool. I think we should read "(Fig. 1; 6, 7).

We apologize for the missing cardiolipin molecule. Figure 1 has been fixed. We have also corrected the reference in the text.

2. Figure 2a: since the TLC aspect is modified according to Figure S4, it is very difficult to discriminate PC from SM and lyso PC, and vice versa. For example, in the text it is stated cells unadapted to POPC + cholesterol (p=1) show only traces of SM, but looking at the figure this seems to be the opposite. I think the full TLC length should be displayed to facilitate interpretation, along with a standard as in Figure S4. In addition, I think the right part of the figure (Diether PC + Cholesterol diet) is currently not mentioned anywhere in the text.

Thank you for highlighting the need to improve clarity in this figure and the results section. Including the full TLCs in the main figure would make it overly cluttered, in our opinion. However, we have included the full TLCs for all conditions shown, along with standards, in Supplementary Figure S2. The primary objective of the TLCs in Figure 2a was to visually demonstrate the reduction in the number of lipid bands as we progress towards more stringent lipid diets, and that internally synthesized lipids (PG and CL) appear after cells have been passaged on a diet of only POPC and cholesterol, but not when passaged on a diet of Diether-PC and cholesterol. The bands for SM and Lyso-PC run close together, and even with adjustments to the aspect ratio, their visual separation remains limited. While a detailed discussion of Lyso-PC is not essential to our conclusions, we included it because it aligns with the idea that acyl chains are scavenged from PC by lipase activity. To enhance the figure's readability, we have more carefully annotated the TLCs to clearly indicate the positions of the key PC and SM bands referenced in the text. Additionally, we have removed the annotation for the Lyso-PC band in the FBS TLC, as its presence in FBS cells is not pertinent to our discussion.

Also, thank you for pointing out that we did not reference the right-hand part of figure 2a. We have now referenced it in the text as follows (line 177):

Following the transfer of cells from FBS or minimal POPC-cholesterol diet to a minimal Diether PC-diet, TLC analysis of lipid extracts over several passages showed the disappearance of cardiolipin and PG, and presence of only two bands corresponding to cholesterol and Diether PC (D.PC; Fig. 2a).

3. Figure S4, related to Figure 2 and 3: I think it is a great idea to give full TLCs results in supplementary materials. However, this figure looks like a lab notebook; labels and terms are not uniform (e.g. std vs standard, Lyso PC vs Lysophospholipid, etc.), there are some cropped labels (... "terol" on TLC #10) and some ambiguities ("cells" refers to mycoplasma mycoides or syn3??), and the figure legend is full of errors (Lipids with a cap I in title, Cardiolipid instead of cardiolipin at line 1, D. POPC repeated twice at last line, lacking abbreviation description such as DOPE, DPPG, etc.). On top of that, essential information is lacking to allow correspondence with TLCs displayed in Figure 2 and 3, or results given in Figure S1, which is critical in the circumstances. A label above each TLC could help figure out which TLC goes with which experiment described in main text. Overall, this makes the interpretation of TLCs placed in main really painful. Please revise, correct, and make labels more uniform.

We apologize for the state of the data presented in figure S4. The presentation of the TLC plots has been standardized, and updated with respect to aesthetics and clarity.

4. Legends of supplementary figures are filled with errors and are lacking abbreviations to clearly understand the figures. Here is a non-exhaustive list of errors that I found by quickly reading them:
 - Figure S1: "specificallyby" lacking a space at line 2; "Deteermined" at line 3; "occurring" instead of "occurring" at line 5
 - Figure S3: "occurring" instead of "occurring" at last line
 - Figure S4: see my previous comment

We greatly apologize for the errors in the supplementary figure text, the entire supplement has been proof read and both text and figures have been corrected for both accuracy and aesthetics.

5. Lines 123 and 408: where are the supp Table S1 and S2? These files seem to be missing from the current manuscript.

We apologize for the missing files, they should have been included with the original submission; we have ensured they have been uploaded and that they are included in this version of the document, along with Table S3 showing the full lipidomics results from figure 2 and 4.

6. Line 128: what are the 28 lipid structures identified by MS? Could you give the list in a supp table?

Thank you for this suggestion. The full list of structures has been added as Supplementary Table 3.

7. Line 148: where are those growth rate results? If they correspond to D. PC in Figure 3f, please specify.

We apologize for the lack of clarity. The manuscript has been updated to clarify the reference to the figure. The correction is here (lines 182-183):

Minimizing the lipidome of *M. mycoides* to two lipids resulted in a two-fold decrease in growth rates (Figure. 3e, D.PC condition compared to POPC condition).

8. From the growth results shown in Figure 3f and Figure S5, it is really difficult to confirm that the cells are really growing in the different media and not just “surviving” for a certain period. For example, in Figure 3f, the results represent a growth rate based on phenol red color change. If we convert this growth rate to a doubling time, this gives a doubling time of ~8 hours ($GR = \ln 2 / DT$), which is greatly lower than the doubling time of *M. mycoides* in SP4 medium (~60 min). If I misinterpreted the y-axis of this figure and it instead represents a medium pH change, then where is the graph showing the correlation between medium pH and OD560nm for their media? Anyhow, the authors should provide raw growth curves of all growth conditions compared in Figure 3f and add a SP4 + FBS control to clarify this.

This is certainly an important point to make clear, and we thank the reviewer for alerting us to the fact that it was not sufficiently emphasized in the manuscript. We are very confident that the cells are growing based on the fact that a) we passage them continuously and if they were not growing cells would be diluted out after several passages b) we get colony forming units when spotting out media from culture flasks and b) we obtain measurable differences in media pH via Phenol Red absorbance that are consistent with growth. Furthermore, new EM images of cells on the Diether-PC + Cholesterol diet demonstrate that there are in fact cells in our cultures (these are cultures that have been passaged for months now). In a separate study with a collaborator we have microcalorimetry data showing that Syn3A cells with a 2-component lipidome generate measurable heat flow, demonstrating growth (we cannot include that data in this manuscript, as it is part of another study). To provide more robust evidence of growth, we have now included, raw growth curves and a media blank (Supplementary Fig. S4a), a calibration curve of media pH vs phenol red absorbance (Supplementary Fig S3a) in the supplementary info as well as a reference growth curve for cells grown on SP4 + FBS (Supplementary Fig. S5 a and b). Furthermore, an experiment comparing CFU growth to OD600 and phenol red growth measures has been added to Supplementary Fig. S5. Showing FBS growth rates, which are around 10-fold higher, in the main figure,

would completely obfuscate differences between the other conditions, without putting in an axis-break, which we prefer not to do. However, we agree that putting these results in context is useful, and we have added the following text to this section of the results, referencing our previous work measuring growth rates of *M. mycoides* and Syn3 on FBS (lines 207-210):

To put these values in context, the growth rates we estimate from all of the defined diets considered in this study are more than 10-fold lower than for cells grown on a complex FBS lipid diet, highlighting the effect of reducing lipid diet complexity on growth⁴⁴.

9. Figure S5, S6, and Figure 5: I am wondering how reliable OD600nm data are for monitoring the growth of *M. mycoides*. Most Mollicutes are too small to be accurately measured at 600nm, and I do not recall any paper using that method to follow the growth of *M. mycoides*. Most studies used phenol red (as the authors also did) or DNA quantification using fluorescence staining as described in Breuer et al. 2019 and Hutchison et al. 2016, for example. In fact, considering previous studies (e.g., Meur et al. 1989), turbidity only appears after prolonged incubation periods (>18h) and is not correlation with viable cell counts but rather indicates the accumulation of cell agglomerates (i.e. dead cells). Can the authors comment on this? I think that a correlation between CFUs and OD600nm data should be made to ensure that measuring OD600nm is a reliable method to follow to growth of *M. mycoides*.

We provided OD600 curves in the supplementary materials as a second measure of growth to corroborate our Phenol Red based estimates in the main figures. We have routinely used OD600 to measure growth curves in *M. mycoides* and Syn3 in SP4 media (e.g. Safronova et al., 2024). The growth rates that we estimate by OD600 are correlated with those from phenol red absorbance across a broad range as demonstrated by figure S3b. We have confirmed this in another study by measuring the heat flow of cells in a microcalorimeter and these values also corroborate our use of OD600 as a measure for *M. mycoides* and Syn3 growth. Under conditions where cells grow to lower densities, we found that phenol red is far more sensitive as a measure of growth, especially for Syn3, which has a lower OD600 absorbance to cell density ratio. Therefore, we have reported phenol red based curves and growth rates. Regarding the aggregation of dead cells, we are confident that we are observing cell growth with each passage based on our response to your previous comment above.

There are many approaches to measuring change in cell density all of which come with some caveats. For example, measuring DNA abundance assumes that there is a constant amount of DNA/cell, which we know from other members of the Syn3 community is not always true. Phenol Red indirectly measures pH change through excretion of metabolites, which also assumes a coupling between pH and cell density. OD assumes a constant relationship between cell absorbance during a given experiment. None of these approaches is perfect. However, given that we observe colony forming units in our cultures, and that these cultures are passaged continuously (still to this date in the lab), we are confident that we are observing growth. We have now added a correlation between phenol red and OD600 growth estimates and colony forming units (Supplementary Fig. S5), which demonstrates a correlation between CFU and both measures of absorbance, but confirms our decision to report Phenol Red based growth for Syn3 in the main figures.

10. Related to my previous point: in Figure S5b, I'm not sure what these units mean (e.g., 0.250, 0.251, 0.252). Am I missing something here? These log axes and units make this graph uninterpretable. If OD600nm is indeed a reliable method to follow the growth of *M. mycoides*, we should see a clear curve using non-transformed data and linear axes. This gives the impression that OD600nm and A562 are correlated only for a very small range of values, and not necessarily during the exponential growth

phase. In Figure S5A, where are these points located on the growth curve? Once again, the transformation of data makes the interpretation difficult. OD600 and A562 could be correlated for a very small range of values (which could be the case here considering the very small A562 variations plotted on the graph) during the stationary or death phase and gives the impression that both methods are reliable to measure growth.

We apologize for the lack of clarity. The scale for figure S5b were 0.25^0 , 0.25^1 , and 0.25^2 , the format of those numbers has been edited for clarity. For more clarity, all growth curves have been plotted on a linear scale in the Supplementary Figures.

11. Line 153: did your lipidomics data confirm that cells only contain 16:0/18:1 diether PC configurations? Could you provide a table of analyzed lipidomics data (same for the 28 lipid structures at line 128)?

Thank you for this suggestion. The full list of lipids for all samples analyzed by shotgun lipidomics has been added as Supplementary Table 3. The lipidomic data shows that 99.9% of the lipidome is composed of cholesterol and 16:0/18:1 D.PC.

12. Figure 4a: I do not see the interest of showing how the JCVI-syn3A genome was segmented and built here. Is there a relation between a certain genome segment and specific results reported in the text? Lines 199 to 204 could be removed or moved in the introduction section.

This is a fair point. We have removed this panel from the figure, and referenced the original Hutchison et al., 2016 paper in which the creation of the Syn3 genome was described.

13. Lines 216-217: "To not bias the growth rates, these cells were removed from the growth rate analysis". How do you remove dead cells from cultures? Do you mean replicates?

We apologize for the lack of clarity. We meant replicates. After revising the manuscript, we realized that this data, while useful supporting information, was not central to the main line of reasoning in the manuscript. Consequently, we moved this data to Supplementary Table S2, along with complementary data for *M. mycoides*, and we have included the following text in the table caption to clarify what we meant:

As a more minimal system, JCVI-Syn3A is a more fragile organism, and not all replicates for a given lipid diet survived the growth protocol. To not bias the growth rates, these replicates were removed from the growth rate analysis. An interesting feature was noticed, however, that a much higher fraction of replicates on diets that could not undergo acyl chain scavenging did not grow compared to replicates on diets that could further illustrating the decreased fitness of cells unable to remodel acyl chain composition.

14. Figure 4b and 4c: once again, given the very low growth rate reported, I would be very interested in seeing the raw growth curves of these conditions, especially since the authors reported that some replicates were also not growing in many conditions. How do these results compare to a JCVI-syn3A grown in SP4 + FBS? Without this comparison, I think the authors should mitigate their statement at lines 226-228.

This is an important point to clarify. The cells grow very slowly and to relatively low densities. The growth curves reflect that, and only provide a rough measure of rate (as exemplified by the large variance

exhibited by the replicates, indicated in the growth rate figures). However, the cells are alive, as evidenced by the fact that we passaged them continuously for >3 passages, get colony forming units from liquid cultures, and TEM imaging reveals presence of intact cells. To address these concerns, raw growth rates curves have been added to supplementary figure S4. Furthermore, an experiment comparing CFU growth to OD600 and phenol red growth rates has been added to figure S5. To put these growth rates in context with cells grown on FBS we have added the following text (lines 284-288):

By comparison, cells grown on FBS, a complex lipid diet, exhibit ~ 10-fold higher growth rates⁴⁴. Although growth is exceptionally slow, the culture can be continuously passaged in batch, and samples taken 24 hours post-inoculation consistently yield colony-forming units, confirming the viability of the cells (Supplementary Fig S5).

15. How many replicates did the authors perform for growth experiments shown in Fig. 3f, 4c, and figure 5? I could not find this information anywhere in the text.

We apologize for leaving that information out; We always considered at least 5 replicates. It has now been added in abbreviated form to the appropriate figure captions, and in full form in Supplementary Table S2, referenced in the “Statistical Analysis” section of Materials and Methods. Figures have also been edited for aesthetics, clarity, and consistency.

16. Figure 5b and 5c: how were these growth rate calculated? Phenol red or OD600?

For figure 5b and 5c growth rates was calculated using phenol red. We apologize for the confusion and have updated the figure title for clarity.

17. Histograms in figures 6 and 5b, c: how many replicates? Do bars and error bars indicate mean + SD, median + SEM, or something else? Please specify.

We apologize for leaving that information out; It has now been added in abbreviated form to the appropriate figure captions, and in full form under the “Statistical Analysis” section of Materials and Methods. Figures have also been edited for aesthetics, clarity, and consistency

18. The information about statistical tests seems to be lacking in the entire manuscript. For example, what test did the authors use in Figure 5b, c, and 6?

We apologize for leaving that information out; It has now been added in abbreviated form to the appropriate figure captions, and in full form under the “Statistical Analysis” section of Materials and Methods. Figures have also been edited for aesthetics, clarity, and consistency

19. Supplementary Figures S2 and S3 are not cited anywhere in the text.

We apologize for the oversight. These were included in an original version of the manuscript, and now they are no longer relevant. They have been removed.

Minor points:

1. Line 40: *Mycoplasma*"s" are a class? Or if you are referring to the genus, italicize and put singular.

We apologize for the lack of clarity; we are referring to the genus *Mycoplasma*. The manuscript has been updated to reflect the correction.

2. Line 43, 369, and 382: italicize "in situ", "in vitro", and "in vivo".

We apologize for the mistake. The manuscript has been updated to reflect the correction.

3. Line 50: add a dot after "subsp" ("subsp.").

We apologize for the mistake. The manuscript has been updated to reflect the correction.

4. Line 51: do not italicize the strain name GM12.

We apologize for the mistake. The manuscript has been updated to reflect the correction.

5. Line 54-55: kind of difficult to read. Please reformulate to clarify.

We apologize for the lack of clarity, the sentence has been reformatted.

6. End of line 91: missing dot at the end of sentence.

We apologize for the mistake. The manuscript has been updated to reflect the correction.

7. Line 109: CL was not defined earlier in the text. Define at line 95 or 97 before using the abbreviation.

We have double checked to make sure all abbreviations are spelled out the first time they are mentioned, and we have added a table with abbreviated chemical names at the beginning of the manuscript. We appreciate the suggestion and apologize for the lack of clarity.

8. Line 111: Define POPC before using the abbreviation.

We have added a section for abbreviated chemical names at the beginning of the manuscript. We appreciate the suggestion and apologize for the lack of clarity.

9. Line 116: Define SM before using the abbreviation. Many lipid abbreviations are not defined before using in the text. Revise the entire manuscript.

We have added a section for abbreviated chemical names at the beginning of the manuscript. We appreciate the suggestion and apologize for the lack of clarity.

10. Figure 2: to facilitate interpretation, please define lipid abbreviations in all Figure legends, including supplemental figures.

We apologize for the lack of clarification in the figure captions, and have updated the manuscript to have full names in all figure captions as well a dedicated section for abbreviations at the start of the manuscript.

11. Line 184: replace "do no" by "do not".

We apologize for the mistake. The manuscript has been updated to reflect the correction.

12. Line 219: there is a comma missing in this sentence.

We apologize for the mistake. The manuscript has been updated to reflect the correction.

13. Line 335-336: italicize "Mesoplasma". I would also cite the original description of Mesoplasma florum L1 to support that point (McCooy et al. 1984).

We apologize for the mistake. The manuscript has been updated to reflect the correction. The supporting publication has also been added as a citation.

14. Line 357 and 359: italicize "E. coli".

We apologize for the mistake. The manuscript has been updated to reflect the correction.

15. Line 392: "t" in "the Saenz" lab is underlined.

We apologize for the mistake. The manuscript has been updated to reflect the correction.

16. Lines 426-427: E0 or Eo? Different between equation and text.

We apologize for the mistake. The manuscript has been updated to reflect the correction.

17. Figure 4c legend: since D. PC comparison is not significant, change "same diets" for "D. PC diet".

We apologize for the mistake. The manuscript has been updated to reflect the correction.

18. Lines 523-549: the font size and type seem to be varying in this paragraph. Same for the liposome preparation paragraph. Please make uniform.

We apologize for the mistake. The manuscript has been updated to reflect the correction.

REVIEWER COMMENTS

Reviewer #1 (Remarks to the Author):

The authors have done a solid job in responding to my comments (and those of the other reviewer), resulting in a significantly improved manuscript. I have no further concerns and I am happy to endorse the publication of the manuscript.

Thank you!

Reviewer #2 (Remarks to the Author):

The revisions by Justice et al substantially improve this interesting manuscript. I now support its publication, with the following comments:

1) I especially appreciate the addition of electron microscopy data and tomography on JCVI-Syn3A. I only wish this type of analysis was done more broadly, e.g. on *M. mycoides* minimal lipid systems, to understand if the intracellular vesicles are a result of the minimal host or the lipids.

2) As the methods make clear now, the diether-PC was quantified using a PC standard. It does not appear that the response factor for PC was calibrated with that for diether-PC. Thus, analysis of diether-PC is not quantitative, but more of an estimate. This does not impact any of the paper's conclusion, but should be noted in the methods and/or caption for Figure 2.

Thank you! Regarding quantification of the diether PC we have modified the text in the Results section as follows (lines 671-675):

Diether PC was quantified using the diester PC standard mentioned above. Although both lipids share the same head group chemistry, differences in ester and ether linkages may cause slight variations in ionization efficiency and mass spectrometric response. As a response factor calibration was not performed, the quantification of diether PC should be regarded as semi-quantitative.

Reviewer #3 (Remarks to the Author):

I thank the authors for revising and correcting their manuscript. I have no further comments.

Thank you!